# Efficient Video Diffusion Models via Content-Frame Motion-Latent Decomposition

**Sihyun Yu**[1*]  **Weili Nie**[2]  **De-An Huang**[2]  **Boyi Li**[2,3]  **Jinwoo Shin**[1]  **Anima Anandkumar**[4]
[1]KAIST   [2]NVIDIA Corporation   [3]UC Berkeley   [4]Caltech

## Abstract

Video diffusion models have recently made great progress in generation quality, but are still limited by the high memory and computational requirements. This is because current video diffusion models often attempt to process high-dimensional videos directly. To tackle this issue, we propose content-motion latent diffusion model (CMD), a novel, efficient extension of pretrained image diffusion models for video generation. Specifically, we propose an autoencoder that succinctly encodes a video as a combination of a content frame (like an image) and a low-dimensional motion latent representation. The former represents the common content, and the latter represents the underlying motion in the video, respectively. We generate the content frame by fine-tuning a pretrained image diffusion model, and we generate the motion latent representation by training a new lightweight diffusion model. A key innovation here is the design of a compact latent space that can directly and efficiently utilize a pretrained image model, which has not been done in previous latent video diffusion models. This leads to considerably better quality generation and reduced computational costs. For instance, CMD can sample a video $7.7\times$ faster than prior approaches by generating a video of $512\times1024$ resolution and length 16 in 3.1 seconds. Moreover, CMD achieves an FVD score of 238.3 on WebVid-10M, 18.5% better than the previous state-of-the-art of 292.4.

## 1 Introduction

Recently, deep generative models have exhibited remarkable success in synthesizing photorealistic and high-resolution images using diffusion models (DMs) (Ho et al., 2020; Nichol & Dhariwal, 2021; Song et al., 2021b; Karras et al., 2022) and even achieving promising results in difficult text-to-image (T2I) generation (Rombach et al., 2022; Saharia et al., 2022; Balaji et al., 2022). Inspired by the success in the image domain, several works have focused on solving a considerably more challenging task of video generation. In particular, they have attempted to design DMs specialized for videos and shown reasonable video generation quality (Ho et al., 2022b; Yang et al., 2022; Yu et al., 2023b; Blattmann et al., 2023). Nevertheless, unlike the image domain, there is still a considerable gap in video quality between generated and real-world videos. This is mainly due to the difficulty of collecting a large training dataset of high-quality videos (Ho et al., 2022b; Ge et al., 2023) and the high dimensionality of video data as cubic arrays, leading to a heavy memory and computational burden (He et al., 2022; Yu et al., 2023b).

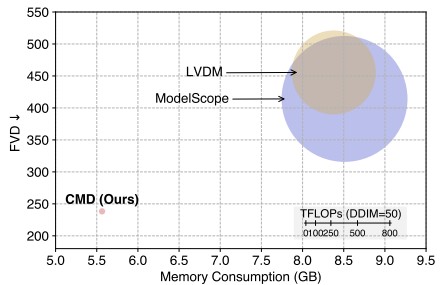

Figure 1: Existing (text-to-)video diffusion models extended from image diffusion models often suffer from computation and memory inefficiency due to extremely high-dimensionality and temporal redundancy of video frames. Compared with these methods, CMD requires ~**16.7**× **less computation with only** ~**66% GPU memory usage in sampling**, while achieving significantly better video generation quality. FLOPs and memory consumption are measured with a single NVIDIA A100 40GB GPU to generate a single video of a resolution $512\times1024$ and length 16.

---

[*]Work done during an internship at NVIDIA. Project page: https://sihyun.me/CMD.

Figure 2: Comparison with (a) the conventional extension of image diffusion models for video generation and (b) our CMD. We mark the newly added parameters as blue. Unlike common approaches that directly add temporal layers in pretrained image diffusion models for extension, CMD encodes each video as an image-like content frame and motion latents, and then fine-tunes a pretrained image diffusion model (*e.g.*, Stable Diffusion (Rombach et al., 2022)) for content frame generation and trains a new lightweight diffusion model (*e.g.*, DiT (Peebles & Xie, 2023)) for motion generation.

To tackle the data collection issue, several video DM approaches leverage pretrained image DMs for video generation (He et al., 2022; Singer et al., 2023; Luo et al., 2023; Ge et al., 2023). Due to the rich visual knowledge already learned from image datasets, the use of image DMs in video generation leads to better generation quality and faster training convergence compared to training a video DM from scratch (An et al., 2023; Blattmann et al., 2023). However, since these video models directly generate high-dimensional videos as cubic arrays, they still entail high memory consumption and computational costs, especially for high-resolution and long videos.

Another line of video DM approaches focuses on alleviating memory and computational inefficiency by first projecting the video into a low-dimensional latent space and then training a DM in the latent space (Yu et al., 2023b). In particular, these approaches consider both the temporal coherency of videos as well as frame-wise compression in video encoding to obtain the maximum efficiency. However, such latent video DMs are only trained on a limited amount of video data and do not incorporate pretrained image models, which limits their video generation quality.

**Our approach.** We address the aforementioned shortcomings by introducing content-motion latent diffusion model (CMD), a memory- and computation-efficient latent video DM that leverages visual knowledge present in pretrained image DMs. CMD is a two-stage framework that first compresses videos to a succinct latent space and then learns the video distribution in this latent space. A key difference compared to existing latent video DMs is the design of a latent space that directly incorporates a pretrained image DM. See Figure 2 for an illustration.

In the first stage, we learn a low-dimensional latent decomposition into a content frame (like an image) and latent motion representation through an autoencoder. Here, we design the content frame as a weighted sum of all frames in a video, where the weights are learned to represent the relative importance of each frame. In the second stage, we model the content frame distribution by fine-tuning a pretrained image DM without adding any new parameters. It allows CMD to leverage the rich visual knowledge in pretrained image DMs. In addition, we design a new lightweight DM to generate motion latent representation conditioned on the given content frame. Such designs avoid us having to deal directly with video arrays, and thus, one can achieve significantly better memory and computation efficiency than prior video DM approaches built on pretrained image DMs.

We highlight the main contributions of this paper below:

- We propose an efficient latent video DM, termed content-motion latent diffusion model (CMD).
- We validate the effectiveness of CMD on popular video generation benchmarks, including UCF-101 (Soomro et al., 2012) and WebVid-10M (Bain et al., 2021). For instance, measured with FVD (Unterthiner et al. 2018; lower is better), our method achieves 238.3 in text-to-video (T2V) generation on WebVid-10M, 18.5% better than the prior state-of-the-art of 292.4.
- We show the memory and computation efficiency of CMD. For instance, to generate a single video of resolution 512×1024 and length 16, CMD only requires 5.56GB memory and 46.83 TFLOPs, while recent Modelscope (Wang et al., 2023a) requires 8.51GB memory and 938.9 TFLOPs, significantly larger than the requirements of CMD (see Figure 1).

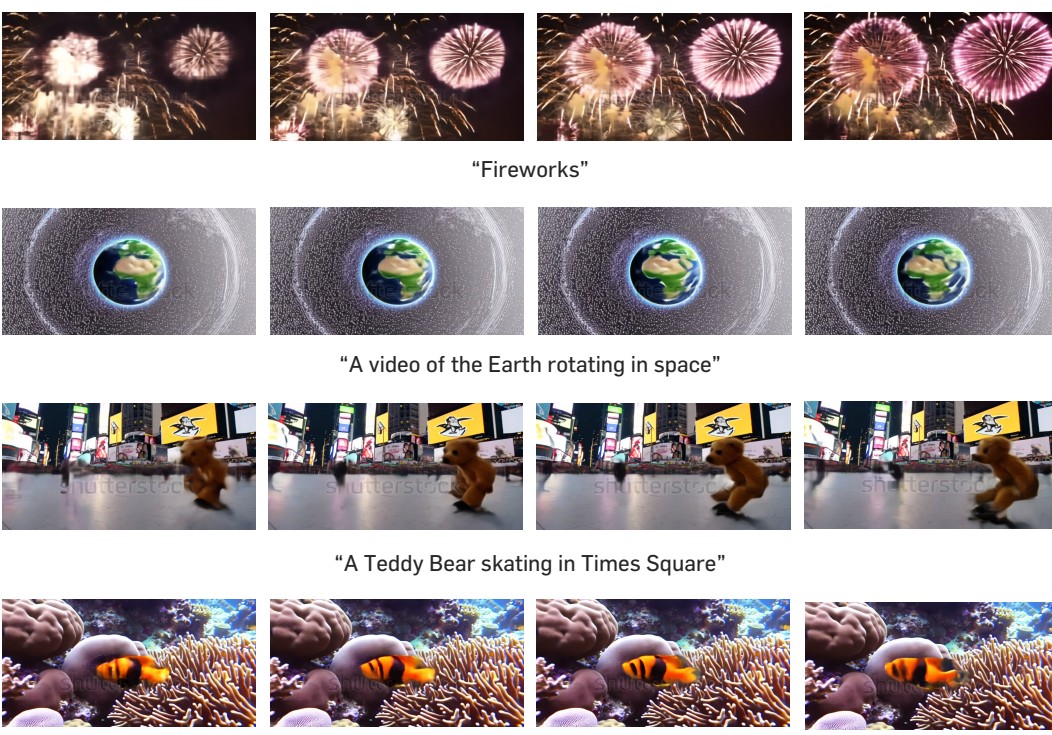

"Fireworks"

"A video of the Earth rotating in space"

"A Teddy Bear skating in Times Square"

"A clownfish swimming through a coral reef"

Figure 3: **512×1024 resolution, 16-frame text-to-video generation results** from our CMD. We visualize video frames with a stride of 5. We provide more examples with different text prompts in Appendix C, as well as their illustrations as video file formats in the supplementary material.

## 2 RELATED WORK

In this section, we provide a brief overview of some of the important relevant literature. For a more extensive discussion with a detailed explanation of other methods, see Appendix A.

**Latent diffusion models.** Diffusion models have suffered from memory and computation inefficiency because they require a large number of iterations in high-dimensional input space for sampling (Ho et al., 2020). To mitigate this issue, several works have considered training diffusion models in a low-dimensional latent space, learned by an autoencoder (Zeng et al., 2022; Xu et al., 2023; Ben Melech Stan et al., 2023). In particular, this approach has shown remarkable success in the image domain (Rombach et al., 2022) to greatly improve efficiency as well as achieve high-quality synthesis results conditioned at a complex text prompt. Similarly, our work aims to design a latent diffusion model for videos (He et al., 2022; Yu et al., 2023b) to alleviate the inefficiencies.

**Video generation.** Numerous works have actively focused on solving the challenging problem of video synthesis. Previously, generative adversarial network (GAN; Goodfellow et al. 2014) based approaches (Gordon & Parde, 2021; Tian et al., 2021; Fox et al., 2021; Munoz et al., 2021; Yu et al., 2022; Skorokhodov et al., 2022; Singer et al., 2023) were proposed to achieve the goal, mostly by extending popular image GAN architectures (Karras et al., 2020). Recently, there have been several works that encode videos as sequences of discrete tokens (van den Oord et al., 2017), where they either generate tokens in an autoregressive manner (Kalchbrenner et al., 2017; Weissenborn et al., 2020; Rakhimov et al., 2020; Yan et al., 2021; Ge et al., 2022) or a non-autoregressive manner (Yu et al., 2023a). In addition, with the success of diffusion models (Ho et al., 2020; Nichol & Dhariwal, 2021) in image generation, recent methods exploit diffusion models for videos (Ho et al., 2022b; Harvey et al., 2022; Yang et al., 2022; Höppe et al., 2022; Singer et al., 2023; Lu et al., 2023), achieving promising results in modeling complex video distribution. Inspired by their success, we also aim to build a new video diffusion model to achieve better video synthesis quality.

**Text-to-video (T2V) generation.** Following the success of text-to-image (T2I) generation (Rombach et al., 2022; Saharia et al., 2022; Balaji et al., 2022), several works have attempted to solve a more challenging task of T2V generation. The main challenge of T2V generation is to resolve the tremendous training costs of diffusion models and the difficulty in collecting large-scale and high-quality video data. Predominant approaches (Ho et al., 2022a; Wang et al., 2023b; An et al., 2023; Blattmann et al., 2023; Ge et al., 2023; He et al., 2022; Singer et al., 2023) have achieved this by fine-tuning pretrained T2I diffusion models by adding temporal layers (*e.g.*, temporal attention and 3D convolution layers) to the 2D U-Net architecture (Saharia et al., 2022). However, they suffer from high memory consumption and computational costs due to the unfavorable increase of input dimension in high-resolution and long videos. To tackle this issue, a few recent works have focused on alternative extension strategies that avoid dealing with entire raw video frames directly (Luo et al., 2023). Specifically, they achieve it by proposing frame-by-frame generation with an additional lightweight diffusion model. However, our extension is based on a latent diffusion model approach to encoding videos as content frames and motion latent representation to reduce the input dimension and learn video diffusion models on such compact latent representation.

## 3 CMD: CONTENT-MOTION LATENT DIFFUSION MODEL

Consider a condition-video pair dataset $\mathcal{D}$, where each sample $(\mathbf{c}, \mathbf{x}^{1:L}) \in \mathcal{D}$ is drawn from an unknown data distribution $p_{\text{data}}(\mathbf{x}^{1:L}, \mathbf{c})$. Here, each $\mathbf{c}$ denotes a condition (*e.g.*, video class or text description) of the corresponding $\mathbf{x}^{1:L}$, and each $\mathbf{x}^{1:L} := (\mathbf{x}^1, \dots, \mathbf{x}^L)$ is a video clip of length $L > 1$ with a resolution $H \times W$, *i.e.*, $\mathbf{x}^\ell \in \mathbb{R}^{C \times H \times W}$ with a channel size $C$. Using $\mathcal{D}$, We aim to learn a conditional model distribution $p_{\text{model}}(\mathbf{x}^{1:L}|\mathbf{c})$ to match the data distribution $p_{\text{data}}(\mathbf{x}^{1:L}|\mathbf{c})$.

Our main idea is to encode each video into an "image-like" content frame and succinct motion latent representation, where pretrained image diffusion models can be used to generate content frames due to the similarity between natural images and content frames. By doing so, rich visual knowledge learned from image data is leveraged for video synthesis, leading to better generation quality as well as reduced training costs. Given content frames, the video generation task thus reduces to designing a motion diffusion model to generate much lower-dimensional motion latent representation.

In the rest of this section, we explain our content-motion latent diffusion model (CMD) in detail. In Section 3.1, we provide an overview of diffusion models. In Section 3.2, we describe our video encoding scheme and design choices of diffusion models for video generation.

### 3.1 DIFFUSION MODELS

The main concept of diffusion models is to learn the target distribution $p_{\text{data}}(\mathbf{x})$ via a gradual denoising process from Gaussian distribution $\mathcal{N}(\mathbf{0_x}, \mathbf{I_x})$ to $p_{\text{data}}(\mathbf{x})$. Specifically, diffusion models learn a *reverse* process $p(\mathbf{x}_{t-1}|\mathbf{x}_t)$ of the pre-defined *forward* process $q(\mathbf{x}_t|\mathbf{x}_0)$ that gradually adds the Gaussian noise starting from $p_{\text{data}}(\mathbf{x})$ for $1 \le t \le T$ with a fixed $T > 0$. Here, for $\mathbf{x}_0 \sim p_{\text{data}}(\mathbf{x})$, $q(\mathbf{x}_t|\mathbf{x}_{t-1})$ can be formalized as $q(\mathbf{x}_t|\mathbf{x}_{t-1}) := \mathcal{N}(\mathbf{x}_t; \alpha_t\mathbf{x}_0, \sigma_t^2\mathbf{I_x})$, where $\sigma_t$ and $\alpha_t := 1 - \sigma_t^2$ are pre-defined hyperparameters with $0 < \sigma_1 < \dots < \sigma_{T-1} < \sigma_T = 1$. If $T$ is sufficiently large, the reverse process $p(\mathbf{x}_{t-1}|\mathbf{x}_t)$ can be also formalized as the following Gaussian distribution:

$$p(\mathbf{x}_{t-1}|\mathbf{x}_t) := \mathcal{N}\Big(\mathbf{x}_{t-1}; \frac{1}{\sqrt{\alpha_t}}\big(\mathbf{x}_t - \frac{\sigma_t^2}{\sqrt{1 - \bar{\alpha}_t}}\boldsymbol{\epsilon}_{\boldsymbol{\theta}}(\mathbf{x}_t, t)\big), \sigma_t^2\mathbf{I_x}\Big), \tag{1}$$

where $\bar{\alpha}_t := \prod_{i=1}^{t}(1-\sigma_i^2)$ for $1 \le t \le T$. Here, $\boldsymbol{\epsilon}_{\boldsymbol{\theta}}(\mathbf{x}_t, t)$ can be trained as a denoising autoencoder parameterized by $\boldsymbol{\theta}$ using the $\boldsymbol{\epsilon}$-prediction objective with a noise $\boldsymbol{\epsilon} \sim \mathcal{N}(\mathbf{0_x}, \mathbf{I_x})$ (Ho et al., 2020):

$$\mathbb{E}_{\mathbf{x}_0, \boldsymbol{\epsilon}, t}\Big[||\boldsymbol{\epsilon} - \boldsymbol{\epsilon}_{\boldsymbol{\theta}}(\mathbf{x}_t, t)||_2^2\Big] \text{ where } \mathbf{x}_t = \sqrt{\bar{\alpha}_t}\mathbf{x}_0 + \sqrt{1 - \bar{\alpha}_t}\boldsymbol{\epsilon}. \tag{2}$$

As the sampling process of diffusion models usually requires a large number of network evaluations $p(\mathbf{x}_{t-1}|\mathbf{x}_t)$ (*e.g.*, 1,000 in DDPM; Ho et al. 2020), their generation cost becomes especially high if one defines diffusion models in the high-dimensional data space. To mitigate this issue, several works have proposed latent diffusion models (Rombach et al., 2022; He et al., 2022): they train the diffusion model in a low-dimensional latent space that encodes the data, thus reducing the computation and memory cost. Inspired by their success, our work follows a similar idea of latent diffusion models to improve both training and sampling efficiency for video synthesis.

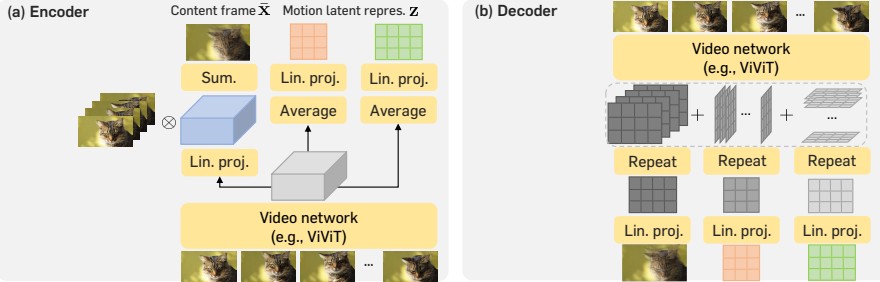

Figure 4: **Illustration of our autoencoder**. Encoder: We compute relative importance of all frames (blue) for a content frame and motion latent representation. Decoder: Using the content frame and motion latent representation, we construct a cubic tensor for video network to reconstruct the video.

## 3.2 EFFICIENT EXTENSION OF IMAGE DIFFUSION MODELS FOR VIDEOS

CMD consists of an autoencoder and two latent diffusion models. First, we train an autoencoder that encodes a video $\mathbf{x}^{1:L}$ as a single content frame $\bar{\mathbf{x}}$ and low-dimensional motion latent representation $\mathbf{z}$. After that, learning the target distribution $p_{\text{data}}(\mathbf{x}^{1:L}|\mathbf{c})$ becomes to learn the following distribution: $p(\bar{\mathbf{x}}, \mathbf{z}|\mathbf{c}) = p(\mathbf{z}|\bar{\mathbf{x}}, \mathbf{c})p(\bar{\mathbf{x}}|\mathbf{c})$. We model each distribution through two diffusion models, where we utilize a pretrained image diffusion model for learning the content frame distribution $p(\bar{\mathbf{x}}|\mathbf{c})$.

**Autoencoder.** We train our autoencoder using a simple reconstruction objective (*e.g.*, $\ell_2$ loss) to encode a video input $\mathbf{x}^{1:L}$. We provide an illustration of the encoder and decoder in Figure 4.

Our encoder $f_{\boldsymbol{\phi}}$ consists of a base network $f_{\boldsymbol{\phi}_B}$ and two heads $f_{\boldsymbol{\phi}_I}, f_{\boldsymbol{\phi}_M}$ for computing $\bar{\mathbf{x}}, \mathbf{z}$ (respectively) with a parameterization $\boldsymbol{\phi} := (\boldsymbol{\phi}_B, \boldsymbol{\phi}_I, \boldsymbol{\phi}_M)$. Here, the base network $f_{\boldsymbol{\phi}_B} : \mathbb{R}^{C \times L \times H \times W} \to \mathbb{R}^{C' \times L \times H' \times W'}$ (with $f_{\boldsymbol{\phi}_B}(\mathbf{x}^{1:L}) = \mathbf{u}$) maps a video $\mathbf{x}^{1:L}$ to hidden feature $\mathbf{u}$ with a channel size $C'$, where we adopt a video transformer (*e.g.*, ViViT; Arnab et al. 2021) as $f_{\boldsymbol{\phi}_B}$. Next, the head $f_{\boldsymbol{\phi}_I} : \mathbb{R}^{C' \times L \times H' \times W'} \to \mathbb{R}^{C \times L \times H \times W}$ returns relative importance among video frames $\mathbf{x}^1, \dots, \mathbf{x}^L$ to compute the content frame $\bar{\mathbf{x}}$. Specifically, we compute $\bar{\mathbf{x}}$ using $f_{\boldsymbol{\phi}_I}$ as:

$$\bar{\mathbf{x}} := \sum_{\ell=1}^{L} \left( \mathbf{x}^{\ell} \otimes \sigma\left( f_{\boldsymbol{\phi}_I}(\mathbf{u}) \right)^{\ell} \right), \tag{3}$$

where $\otimes$ denotes an element-wise product and $\sigma(\cdot)$ is a softmax function across the temporal axis. Consequently, the content frame $\bar{\mathbf{x}}$ has the same dimension with each frame and lies in the space of interpolating consecutive frames, thus looking very similar to them (see Figure 7).

For motion latent representation $\mathbf{z}$, we design it as a concatenation of two latents, *i.e.*, $\mathbf{z} = (\mathbf{z}_{\text{x}}, \mathbf{z}_{\text{y}})$ with $\mathbf{z}_{\text{x}} \in \mathbb{R}^{D \times L \times H'}$ and $\mathbf{z}_{\text{y}} \in \mathbb{R}^{D \times L \times W'}$, where $\mathbf{z}_{\text{x}}, \mathbf{z}_{\text{y}}$ are computed from $\mathbf{u}$ using $f_{\boldsymbol{\phi}_M}$ as follows:

$$(\mathbf{z}_{\text{x}}, \mathbf{z}_{\text{y}}) := \left( f_{\boldsymbol{\phi}_M}(\bar{\mathbf{u}}_{\text{x}}), f_{\boldsymbol{\phi}_M}(\bar{\mathbf{u}}_{\text{y}}) \right). \tag{4}$$

Here, $\bar{\mathbf{u}}_{\text{x}} \in \mathbb{R}^{C' \times L \times H'}, \bar{\mathbf{u}}_{\text{y}} \in \mathbb{R}^{C' \times L \times W'}$ are two projected tensors of $\mathbf{u}$ by simply averaging across x-axis and y-axis, respectively, and $f_{\boldsymbol{\phi}_M}$ is a $1 \times 1$ convolutional layer that maps an input tensor from a channel size $C'$ to $D$. Such a 2D-projection-based motion encoding is motivated by recent triplane video encoding (Kim et al., 2022; Yu et al., 2023b) that project videos to each x, y, t axis.

Similarly, we design a decoder network $g_{\boldsymbol{\psi}}$ as two embedding layers $g_{\boldsymbol{\psi}_I}, g_{\boldsymbol{\psi}_M}$ for $\bar{\mathbf{x}}, \mathbf{z}$ (respectively) and a video network $g_{\boldsymbol{\psi}_B}$ that returns the reconstruction of $\mathbf{x}^{1:L}$ from the outputs of $g_{\boldsymbol{\psi}_I}, g_{\boldsymbol{\psi}_M}$. Specifically, $g_{\boldsymbol{\psi}_I}, g_{\boldsymbol{\psi}_M}$ maps $\bar{\mathbf{x}}, \mathbf{z}$ to have the same channel size $C'$:

$$\mathbf{v}^{\text{t}} := g_{\boldsymbol{\psi}_I}(\bar{\mathbf{x}}) \in \mathbb{R}^{C' \times H' \times W'}, \quad \mathbf{v}^{\text{x}} := g_{\boldsymbol{\psi}_M}(\mathbf{z}_{\text{x}}) \in \mathbb{R}^{C' \times L \times H'}, \quad \mathbf{v}^{\text{y}} := g_{\boldsymbol{\psi}_M}(\mathbf{z}_{\text{y}}) \in \mathbb{R}^{C' \times L \times W'}, \tag{5}$$

where we denote $\mathbf{v}^{\text{t}} = [v_{hw}^{\text{t}}], \mathbf{v}^{\text{x}} = [v_{\ell h}^{\text{x}}], \mathbf{v}^{\text{y}} = [v_{\ell w}^{\text{y}}]$ with $v_{hw}^{\text{t}}, v_{\ell h}^{\text{x}}, v_{\ell w}^{\text{y}} \in \mathbb{R}^{C'}$ for $\ell \in [1, L], h \in [1, H'], w \in [1, W']$. After that, we compute the input of a video network $g_{\boldsymbol{\psi}_B}$, denoted by $\mathbf{v} = [v_{\ell hw}] \in \mathbb{R}^{C' \times L \times H' \times W'}$, by taking the sum of the corresponding vectors of each $\mathbf{v}^{\text{t}}, \mathbf{v}^{\text{x}}, \mathbf{v}^{\text{y}}$, namely:

$$v_{\ell hw} = v_{hw}^{\text{t}} + v_{\ell h}^{\text{x}} + v_{\ell w}^{\text{y}} \text{ for } 1 \le \ell \le L, \ 1 \le h \le H', \ 1 \le w \le W', \tag{6}$$

and then $\mathbf{v}$ is passed to $g_{\boldsymbol{\psi}_B} : \mathbb{R}^{C' \times L \times H' \times W'} \to \mathbb{R}^{C \times L \times H \times W}$ to reconstruct the input video $\mathbf{x}^{1:L}$. For $g_{\boldsymbol{\psi}_B}$, we use the same video transformer architecture as $f_{\boldsymbol{\phi}_B}$.

**Content frame diffusion model.** Recall that the content frame $\bar{\mathbf{x}}$ is computed as a weighted sum of video frames $\mathbf{x}^1, \ldots, \mathbf{x}^L$ and thus it resembles natural images. Hence, for training the content frame diffusion model to learn $p(\bar{\mathbf{x}}|\mathbf{c})$, we directly fine-tune the pretrained image diffusion model $\boldsymbol{\epsilon}_{\boldsymbol{\theta}_I}(\mathbf{x}_t, \mathbf{c}, t)$ without adding additional parameters. In particular, we use condition-content-frame pairs $(\mathbf{c}, \bar{\mathbf{x}})$ computed from the dataset $\mathcal{D}$ and use the denoising objective for fine-tuning:

$$\mathbb{E}_{\bar{\mathbf{x}}_0, \boldsymbol{\epsilon}, t}\left[||\boldsymbol{\epsilon} - \boldsymbol{\epsilon}_{\boldsymbol{\theta}_I}(\bar{\mathbf{x}}_t, \mathbf{c}, t)||_2^2\right] \text{ where } \bar{\mathbf{x}}_t = \sqrt{\bar{\alpha}_t}\bar{\mathbf{x}}_0 + \sqrt{1 - \bar{\alpha}_t}\boldsymbol{\epsilon}. \tag{7}$$

Note that this fine-tuning is memory-efficient since it does not increase input dimension, and it can be trained efficiently due to the small gap between content frames and natural images.

**Motion diffusion model.** To learn the conditional distribution $p(\mathbf{z}|\bar{\mathbf{x}}, \mathbf{c})$, we train a lightweight diffusion model $\boldsymbol{\epsilon}_{\boldsymbol{\theta}_M}(\mathbf{z}_t, \mathbf{c}, \bar{\mathbf{x}}, t)$. For the network architecture, we exploit DiT (Peebles & Xie, 2023), a recently proposed Vision Transformer (ViT) backbone (Dosovitskiy et al., 2020) for diffusion models, due to its better performance and efficiency. Accordingly, for a denoising target $\mathbf{z}_t$, we pass it to the model as a sequence of patch embeddings. Next, for an input condition $\mathbf{c}$, we follow the same conditioning scheme of the original DiT that passes it through the AdaIN layers (Huang & Belongie, 2017). For the conditioning content frame $\bar{\mathbf{x}}$, rather than passing it through the AdaIN layers, we feed it as input-level patch embeddings like $\mathbf{z}_t$ to provide "dense conditions" to the model for predicting motion latent representation $\mathbf{z}$ (see Figure 2). Using these inputs, we train the model via the denoising objective:

$$\mathbb{E}_{\mathbf{z}_0, \boldsymbol{\epsilon}, t}\left[||\boldsymbol{\epsilon} - \boldsymbol{\epsilon}_{\boldsymbol{\theta}_M}(\mathbf{z}_t, \mathbf{c}, \bar{\mathbf{x}}, t)||_2^2\right] \text{ where } \mathbf{z}_t = \sqrt{\bar{\alpha}_t}\mathbf{z}_0 + \sqrt{1 - \bar{\alpha}_t}\boldsymbol{\epsilon}. \tag{8}$$

We observe that a lightweight model can quickly converge to well-predicting motion latent representation $\mathbf{z}$, mainly due to two factors: (a) the rich information provided by the conditions $(\mathbf{c}, \bar{\mathbf{x}})$, and (b) the low dimensionality of motion latent representation $\mathbf{z}$. Moreover, one can use a larger patch size for $\bar{\mathbf{x}}$ (condition) than $\mathbf{z}$ (prediction target) to reduce the total sequence length of input patches to the DiT network, thus further decreasing the computational cost (see Section 4.3).

## 4 EXPERIMENTS

In Section 4.1, we provide setups for our experiments. In Section 4.2, we present the main results, including qualitative results of visualizing generated videos. Finally, in Section 4.3, we conduct extensive analysis to validate the effect of each component as well as to show the efficiency of CMD in various aspects, compared with previous text-to-video generation methods.

### 4.1 SETUPS

**Datasets.** We mainly consider UCF-101 (Soomro et al., 2012) and WebVid-10M (Bain et al., 2021) for the evaluation. We also use MSR-VTT (Xu et al., 2016) for a zero-shot evaluation of the text-to-video models. For model training, we use only train split and exclude test (or validation) sets for all datasets. We provide more details, including how they are preprocessed in Appendix B.1.

**Baselines.** For class-conditional (non-zero-shot) generation on UCF-101, we consider recent DI-GAN (Yu et al., 2022), TATS (Ge et al., 2022), CogVideo (Hong et al., 2023), Make-A-Video (Singer et al., 2023), and MAGVIT (Yu et al., 2023a) as baselines. For zero-shot evaluations, we compare with recent CogVideo, LVDM (He et al., 2022), ModelScope (Wang et al., 2023a), VideoLDM (Blattmann et al., 2023), VideoFactory (Wang et al., 2023b), PYoCo (Ge et al., 2023), GODIVA (Wu et al., 2021), and NÜWA (Wu et al., 2022). See Appendix B.2 for more details.

**Training details.** In all experiments, videos are clipped to 16 frames for both training and evaluation. For a video autoencoder, we use TimeSFormer (Bertasius et al., 2021) as a backbone. For the content frame diffusion model, we use pretrained Stable Diffusion (SD) 1.5 and 2.1-base (Rombach et al., 2022), where each video frame is first encoded by SD image autoencoder into a latent frame with an $8\times$ downsampling ratio and output channel size $C = 4$. For the motion diffusion model, we use DiT-L/2 (for UCF-101) and DiT-XL/2 (for WebVid-10M) as in the original DiT paper (Peebles & Xie, 2023), where "L" and "XL" specify the model sizes and "2" denotes patch size of $2\times2$ when converting input into a sequence of patches. We provide all other details in Appendix B.3.

Table 1: **Class-conditional video generation on UCF-101.** # denotes the model also uses the test split for training. ↓ indicates lower values are better. Bolds indicate the best results, and we mark our method by blue. We mark the method by * if the score is evaluated with 10,000 real data and generated videos, otherwise we use 2,048 videos. For a zero-shot setup, we report the dataset size used for training.

| Method | Zero-shot | FVD ↓ |
|---|---|---|
| DIGAN[#] | No | 465±12 |
| TATS | No | 332±18 |
| CogVideo | No | 305 |
| VideoFusion | No | 173 |
| **CMD (Ours)** | **No** | **107±9** |
| Make-A-Video* | No | 367 |
| MAGVIT* | No | 76±2 |
| **CMD (Ours)*** | **No** | **73±2** |
| VideoFactory | Yes (130M) | 410 |
| PYoCo | Yes (22.5M) | **355** |
| CogVideo | Yes (5.4M) | 702 |
| LVDM | Yes (10.7M) | 642 |
| ModelScope | Yes (10.7M) | 640 |
| VideoLDM | Yes (10.7M) | 551 |
| VideoGen | Yes (10.7M) | 554 |
| **CMD (Ours)** | **Yes (10.7M)** | **504** |

Table 2: **T2V generation on MSR-VTT.** ↑ indicates higher scores are better. Bolds indicate the best results, and we mark our method by blue. We report the dataset size. * denotes LAION-5B (Schuhmann et al., 2022) is jointly used.

| Method | Zero-shot | CLIPSIM ↑ |
|---|---|---|
| GODIVA | No | 0.2402 |
| NÜWA | No | 0.2409 |
| VideoFactory | Yes (130M) | 0.3005 |
| Make-A-Video | Yes (100M) | **0.3049** |
| CogVideo | Yes (5.4M) | 0.2631 |
| LVDM | Yes (10.7M) | 0.2381 |
| VideoLDM | Yes (10.7M) | 0.2929 |
| ModelScope* | Yes (10.7M) | **0.2930** |
| **CMD (Ours)** | **Yes (10.7M)** | 0.2894 |

Table 3: **T2V generation on WebVid-10M.** ↓ and ↑ indicate lower and higher scores are better, respectively. Bolds indicate the best results, and we mark our method by blue. cfg denotes classifier-free guidance scale.

| Method | FVD ↓ | CLIPSIM ↑ |
|---|---|---|
| LVDM | 455.5 | 0.2751 |
| ModelScope | 414.1 | 0.3000 |
| VideoFactory | 292.4 | **0.3070** |
| **CMD (Ours); cfg=9.0** | **238.3** | 0.3020 |

**Metrics.** Following the experimental setup in recent representative video generation literature (Skorokhodov et al., 2022; Yu et al., 2023a), we mainly use Fréchet video distance (FVD; Unterthiner et al. 2018, lower is better) for evaluation. To measure text-video alignment, we additionally measure CLIPSIM (Wu et al. 2021, higher is better) and compare the values with the baselines. We provide more details of evaluation metrics and how they are computed in Appendix B.4.

## 4.2 MAIN RESULTS

**Qualitative results.** We visualize several text-to-video generation results from CMD in Figure 3. As shown in this figure, generated videos contain the detailed motion and contents provided by text prompts and achieve great temporal coherency, leading to a smooth video transition. In particular, the background is preserved well between different video frames in the generated video with the prompt. For instance, "A Teddy Bear Skating in Times Square" maintains details of Times Square well across different video frames. Note that each frame has a resolution of 512×1024, where we achieve such a high-resolution video generation without requiring any spatiotemporal upsamplers. We provide more qualitative results with other text prompts in Appendix C.

**Quantitative results.** Table 1 provides the non-zero-shot generation result on UCF-101 by training all models from scratch on UCF-101 (including the content frame diffusion model). As shown in this table, CMD outperforms all other video generation methods, indicating our framework design itself is an effective video generation method irrespective of the exploitation of pretrained image diffusion models. Moreover, we consider text-to-video generation by training CMD on WebVid-10M with the pretrained SD backbone fine-tuned for content frame generation. As shown in Table 1 and 3, our model shows better FVD scores than previous approaches if the same amount of data is used. Moreover, our model achieves comparable or even better CLIPSIM scores, compared with state-of-the-art as shown in Table 2 and 3, indicating a good text-video alignment. CMD shows a slightly worse CLIPSIM score than ModelScope and VideoLDM on MSR-VTT, but note that our model (1.6B) is ∼1.9× smaller than VideoLDM (3.1B). Moreover, ModelScope *jointly* trains on 5 billion image-text pairs along with video data to avoid catastrophic forgetting, in contrast to CMD that does not use any image data for training once provided pretrained image diffusion models.

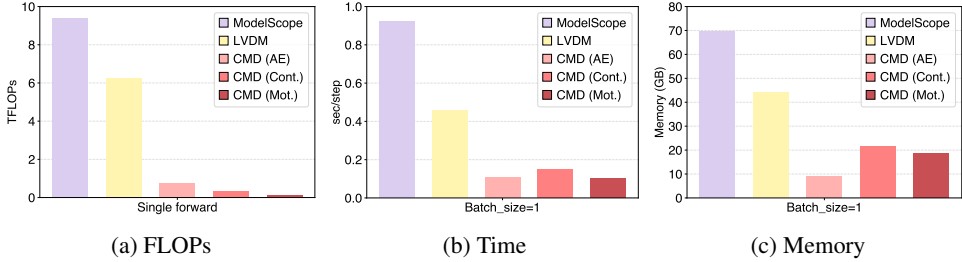

(a) FLOPs                    (b) Time                    (c) Memory

Figure 5: **Training efficiency.** (a) FLOPs, (b) sec/step, and (c) memory (GB) of different methods that are trained on 16-frame videos with resolution of $512 \times 512$ and batch size of 1. All values are measured with a single NVIDIA A100 80GB GPU with mixed precision. For a fair comparison, we do not apply gradient checkpointing for all models. See Appendix D for details.

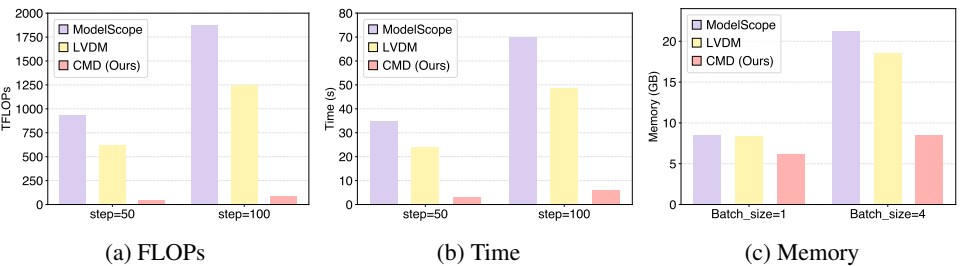

(a) FLOPs                    (b) Time                    (c) Memory

Figure 6: **Sampling efficiency.** (a) FLOPs, (b) time (s), and (c) memory (GB) of different methods that sample a 16-frame video with resolution of $512 \times 1024$ (*i.e.*, batch size = 1 by default). All values are measured with a single NVIDIA A100 40GB GPU with mixed precision. Note that we exclude the cost of Stable Diffusion decoder for all measurements. See Appendix D for details.

## 4.3 ANALYSIS

**Training efficiency.** Figure 5 summarizes the computation (floating point operations; FLOPs), time, and memory consumption in training each component of CMD and compares the values with other public text-to-video diffusion models. As shown in these plots, all components of CMD require less memory and computation for training due to the decomposition of videos as two low-dimensional latent variables (content frame and motion latent representation). Notably, CMD shows significantly fewer FLOPs than prior methods: the bottleneck is in the autoencoder (0.77 TFLOPs) and is $\sim 12\times$ more efficient than 9.41 TFLOPs of ModelScope. Note that if one sums up the FLOPs or training time of all three components in CMD, they are still significantly better than existing text-to-video diffusion models. We also note that the training of content frame diffusion models and motion diffusion models can be done in parallel. Thus, the training efficiency (in terms of time) can be further boosted. We also provide a model parameter size comparison in Appendix D.

**Sampling efficiency.** Figure 6 reports FLOPs, time, and memory consumption to sample videos. As shown in Figure 6a, existing text-to-video diffusion models require tremendous computations for sampling since they directly input videos as high-dimensional cubic arrays. In particular, they overlook common contents in video frames (*e.g.*, static background), and accordingly, many spatial layer operations (*e.g.*, 2D convolutions) become unfavorably redundant and tremendous. However, CMD avoids dealing with giant cubic arrays, and thus, redundant operations are significantly reduced, resulting in a computation-efficient video generation framework. The sampling efficiency is also reflected in sampling time (Figure 6b); CMD only requires $\sim 3$ seconds with a DDIM sampler (Song et al., 2021a) using 50 steps, which is $10\times$ faster than existing text-to-video diffusion models.

Not only improving computation efficiency, our method also exhibits great memory efficiency compared with existing methods due to the significantly reduced input dimension. Note that the improvement becomes more significant if the models sample multiple videos at once (*i.e.*, a batch size larger than 1) because, in that case, the memory bottleneck mainly stems from the computation of intermediate features for sampling rather than the memory allocation of the model parameters. For instance, as shown in Figure 6c, our model uses about 8.6GB GPU memory to generate 4 videos in parallel, $2.5\times$ less consumption than the recent ModelScope model that requires more than 20GB.

Table 4: **Ablation studies.** (a) FVD on UCF-101 to evaluate each component. Reconstruction: FVD between real videos and their reconstructions from our autoencoder. Motion prediction: FVD between real videos and predicted videos with the motion diffusion model conditioning on (ground-truth) content frames encoded by our autoencoder. Content generation: performance of CMD, where content frames are generated by our content frame diffusion model. (b) R-FVD of autoencoders on WebVid-10M with different channel sizes $D$, video lengths $L$, and the usage of weighted sum or not. (c) FVD of motion diffusion models on UCF-101 with different content frame patch sizes.

(a) Performance of each component

| Task | FVD |
|---|---|
| Reconstruction | 7.72 |
| Motion prediction | 19.5 |
| Content generation | 73.1 |

(b) Autoencoder

| $D$ | $L$ | Weight. | R-FVD |
|---|---|---|---|
| 16 | 16 | ✓ | 56.8 |
| 8 | 16 | ✓ | 69.5 |
| 8 | 16 | ✗ | 76.1 |
| 8 | 24 | ✓ | 81.3 |

(c) Motion diffusion

| Config. | $\bar{\mathbf{x}}$ patch. | FVD |
|---|---|---|
| DiT-L/2 | 16 | 40.4 |
| DiT-L/2 | 8 | 32.9 |
| DiT-L/2 | 4 | 19.5 |

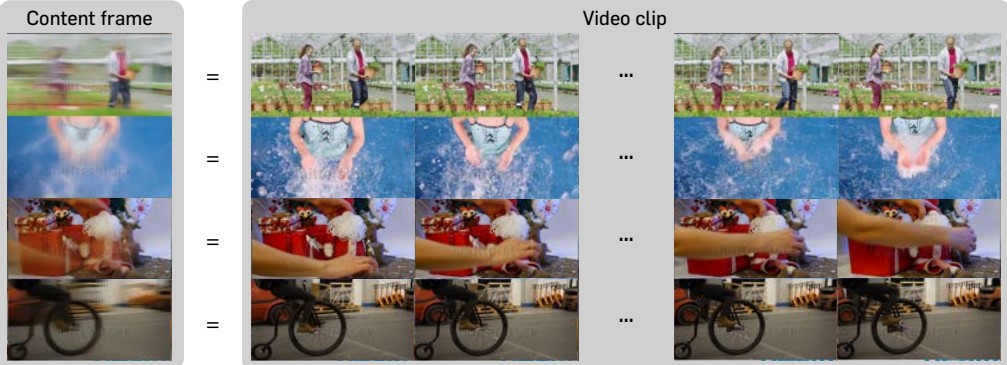

Figure 7: **Content frame visualization** with the corresponding video frames.

**Ablation studies.** In Table 4a, we report the FVD values by using only some of the components in CMD. As shown in this table, each module in CMD shows reasonable performance, which validates our design choices for the overall framework. Moreover, in Table 4b, we analyze the performance of the autoencoder under various setups; one can observe that the use of weighted sum in content frame design helps to achieve better reconstruction, and our autoencoder can encode videos with a longer length than 16 (*e.g.*, $L$ =24) with reasonable quality as well. Finally, Table 4c shows that motion diffusion models exhibit a reasonable performance with large patch sizes, so one can control the tradeoff between computation efficiency and memory efficiency by adjusting the patch size.

**Content frame visualization.** Figure 7 visualizes videos in WebVid-10M and the corresponding content frames. As shown in this figure, the content frames resemble the original video frames, *i.e.*, in the content frames, the background (*e.g.*, buildings) and objects (*e.g.*, a bicycle) appear similarly to the video frames. Moreover, one can observe that only the region with moving objects is corrupted, *e.g.*, for the content frame of a video with a moving arm, an area where the arm appears is corrupted.

## 5 CONCLUSION

We proposed CMD, an efficient extension scheme of the image diffusion model for video generation. Our key idea is based on proposing a new encoding scheme that represents videos as content frames and succinct motion latents to improve computation and memory efficiency. We hope our method will facilitate lots of intriguing directions for efficient large-scale video generation methods.

**Limitation and future work.** In this work, we primarily focused on generating a video of a fixed length (*e.g.*, $L = 16$). One of the interesting future directions would be extending our method for long video synthesis, similar to PVDM (Yu et al., 2023b) which considers clip-by-clip generation. Another interesting direction is to develop a better form of content frame and motion latents to encode video with higher quality but still enable exploiting pretrained image diffusion models. We provide a more detailed discussion of limitation and future work in Appendix I.

## ETHICS STATEMENT

We believe CMD can provide a positive impact in real-world scenarios related to content-creation applications. Since CMD can instantly synthesize videos from arbitrary user text prompts, it can save time for designers (Villegas et al., 2023) who want to generate new content by providing them with an initial shape of such desired result. Moreover, given that the success of large text-to-image generation models (Rombach et al., 2022; Saharia et al., 2022; Balaji et al., 2022) has facilitated intriguing applications such as image editing (Brooks et al., 2023; Kim et al., 2023; Meng et al., 2022) and personalized generation (Ruiz et al., 2023; Gal et al., 2023), we expect developing a large-scale video generation framework will promote similar applications in the video domain as well (Molad et al., 2023).

In contrast, there also exists some potential negative impact of developing a large-scale generation framework to generate sensitive and malicious content, *e.g.*, DeepFake (Guera & Delp, 2018), as discussed by some recent large-scale video generation works (Villegas et al., 2023). Although generated videos from CMD are relatively short and the frame quality is yet distinguishable from real-world videos, one should be aware of this issue and keep considering to develop a safe video generation framework in the future.

## REPRODUCIBILITY STATEMENT

We provide implementation details (*e.g.*, hyperparameter, model, and optimizer) and experiment setups (*e.g.*, how the metrics are computed) in Section 4 and Appendix B.

## ACKNOWLEDGEMENTS

SY thanks Subin Kim, Jaehyun Nam, Jihoon Tack, and anonymous reviewers for their helpful feedback on the early version of the manuscript. SY also acknowledges Seung Wook Kim for helping with text-to-video model training and thanks Yewon Kim for the valuable feedback on the design of the project page. For SY and JS: This work was supported by Institute of Information & communications Technology Planning & Evaluation (IITP) grant funded by the Korea government (MSIT) (No.2019-0-00075, Artificial Intelligence Graduate School Program (KAIST); No.2021-0-02068, Artificial Intelligence Innovation Hub; No.2022-0-00959).

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

## A    MORE DISCUSSION ON RELATED WORK

There are several recent works that have some similarities to CMD. In what follows, we discuss the relevant work and the differences with our method in detail.

**Motion-content decomposition.** CMD is similar to many previous video GANs that generate videos via motion-content decomposition (Villegas et al., 2017; Hsieh et al., 2018; Tulyakov et al., 2018; Tian et al., 2021; Munoz et al., 2021; Yu et al., 2022; Skorokhodov et al., 2022). To achieve generation with motion-content controllability, they sample random content and motion vectors and use them to generate overall style and underlying motions. We take a similar approach using pretrained image diffusion models by training a video autoencoder that decomposes video as a content frame (similar to content vectors) and motion latent representation. Here, note that CMD does not strictly decompose motion and contents in a given video, but the scheme is similar at a high level. Moreover, CMD shares a similarity to several diffusion models for video generation (Guo et al., 2023; Jiang et al., 2023; Ni et al., 2023). However, they have not achieved (a) an efficient generative modeling and (b) exploitation of pretrained image diffusion models simultaneously. Specifically, (Guo et al., 2023) exploits pretrained image diffusion models, but it deals with videos as cubic tensors, (Jiang et al., 2023; Ni et al., 2023) do not use pretrained image diffusion models, and (Ni et al., 2023) uses flow for motion encoding, which is a high-dimensional cubic tensor as well.

**Video prediction.** Our work also has a relationship between video prediction models that forecast future video frames given at previous frames as input (Srivastava et al., 2015; Finn et al., 2016; Denton & Birodkar, 2017; Babaeizadeh et al., 2018; Denton & Fergus, 2018; Lee et al., 2018; Villegas et al., 2019; Kumar et al., 2020; Franceschi et al., 2020; Luc et al., 2020; Lee et al., 2021; Seo et al., 2022). Similar to our method, some video prediction methods predict video frames as low-dimensional latent space rather than raw pixel space to increase the window size and achieve efficiency. The main difference between our method and these approaches is in the input condition: our method provides a content frame as an input, whereas they provide initial frames as an input.

**Difference with PVDM.** Our model shares a similarity to recent latent diffusion models for videos (He et al., 2022; Zhou et al., 2022; Yu et al., 2023b). In particular, our method is quite similar to PVDM (Yu et al., 2023b); this work also proposes a latent video diffusion model to target unconditional video generation based on proposing a video encoding scheme to decompose them as triplane latents. In this work, each latent in triplane latents lies in a different space from the video frames. Thus, it is difficult to exploit pretrained image models. Different from this work, our primary focus is on conditional video generation, and we introduce the "content frames" concept to exploit pretrained image diffusion while avoiding handling giant cubic video tensors.

**Difference with VideoFusion.** Our work shares a similarity to VideoFusion (Luo et al., 2023). Unlike conventional approaches that add temporal layers to the image diffusion models for achieving T2V generation, VideoFusion also considers the training of an additional diffusion model in addition to the pretrained image model for generating videos. However, in contrast to CMD, their primary focus is not on achieving dimension reduction for improving computation and memory efficiency.

**Discussion with concurrent works.** Show-1 (Zhang et al., 2023) also considers an efficient T2V generation via a mixture of pixel-level image diffusion models and latent image diffusion models. LAVIE (Wang et al., 2023c) considers video generation using cascaded latent diffusion models. VideoDirectorGPT (Lin et al., 2023) proposes to combine large language models to generate text prompts for longer video generation. Li et al. (2023b) considers controllable video generation conditioned on a given image and the motion direction. Text2Video-Zero (Khachatryan et al., 2023) considers zero-shot video generation from a pretrained T2I model without any video data. However, neither of them considers the temporal coherency of videos for improving efficiency.

## B IMPLEMENTATION DETAILS

### B.1 DATASETS

**UCF-101** (Soomro et al., 2012) is a human video dataset that includes 101 different types of human actions. Each video consists of frames with 320×240 resolution with varying video lengths. The dataset contains 13,320 videos in total; 9,537 videos are in the train split, and the rest of them are in the test split. We only use train split for training and evaluation of the model, following the experimental setup used in recent video generation methods (Yu et al., 2023a; Singer et al., 2023). We resized all video frames to 64×64 resolution frames and clipped them to a video length of 16. For the zero-shot evaluation, we resize each video to 512×512 resolution.

**WebVid-10M** is a dataset that consists of 10,727,607 text-video pairs as training split. The dataset also contains a validation split that is composed of 5,000 text-video pairs. We use train split for training the model and use a validation set for evaluation. Since some videos in the training split are not available, we exclude these videos from training. We resize all video frames to 512×1024 resolution and clip each video into a length of 16.

**MSR-VTT** is a dataset consisting of 10,000 videos and corresponding captions. We use test split for zero-shot evaluation of our method, which contains 2,000 videos and corresponding text captions. We only use text captions to measure the alignment between text prompts and generated videos.

### B.2 BASELINES

In what follows, we explain the main idea of baseline methods that we used for the evaluation.

- **DIGAN** (Yu et al., 2022) presents a video GAN based on adapting the concept of implicit neural representations (or neural fields) into the generator.
- **TATS** (Ge et al., 2022) presents a video autoencoder by extending an image autoencoder in VQGAN (Esser et al., 2021) and trains an autoregressive Transformer for a generation.
- **CogVideo** (Hong et al., 2023) presents a large-scale autoregressive Transformer for video generation extended from a pretrained model for images.
- **Make-A-Video** (Singer et al., 2023) presents a method to achieve text-to-video generation without having text-video pairs but only with text-image pairs and video data.
- **VideoFusion** (Luo et al., 2023) proposes a new extension scheme of pretrained image diffusion models for video generation by training an additional diffusion model that achieves frame-by-frame generation from intermediate noises from pretrained image diffusion models.
- **MAGVIT** (Yu et al., 2023b) proposes a non-autoregressive Transformer for videos, based on extending a non-autoregressive Transformer for images, MaskGiT (Chang et al., 2022).
- **VideoFactory** (Wang et al., 2023b) proposes a new swapped cross-attention for better video diffusion models and introduces a large-scale text-video dataset.
- **PYoCo** (Ge et al., 2023) explores noise prior to extend image diffusion models for better video generation, instead of starting from i.i.d. Gaussian noises.
- **LVDM** (He et al., 2022) extends latent image diffusion models by modeling video distribution in spatiotemporally downsampled latent space.
- **ModelScope** (Wang et al., 2023a) trains a latent video diffusion model from a pretrained text-to-image diffusion model by adding several temporal layers.
- **VideoLDM** (Blattmann et al., 2023) proposes an extension scheme from text-to-image diffusion models to text-to-video diffusion models by adding temporal layers in pretrained diffusion models and pretrained image autoencoder.
- **VideoGen** (Li et al., 2023a) generates video from a given image and text prompts by generating motion latent representation inspired by flow-based temporal upsampling.
- **GOVIDA** (Wu et al., 2021) trains an autoregressive Transformer with large-scale text-video pair datasets, HowTo100M (Miech et al., 2019).
- **NÜWA** (Wu et al., 2022) extends the approaches in GODIVA to multi-modality, including images, text, and videos.

### B.3 TRAINING DETAILS

**Autoencoder.** In all experiments, we use the Adam optimizer (Kingma & Ba, 2015) with a learning rate of 1e-5, $(\beta_1, \beta_2) = (0.5, 0.9)$ without weight decay. We use 8 NVIDIA A100 80GB GPUs for training with a batch size of 24, and it takes ~1 week for the convergence with 1024×512 resolution videos. For backbone networks, we use TimeSFormer (Bertasius et al., 2021) following PVDM (Yu et al., 2023b), and we use a single convolutional layer as pre- and post-layers at the end of the encoder and at the beginning of the decoder. We provide other hyperparameters related to the model configurations in Table 5:

Table 5: Hyperparameters related to our autoencoder. Hidden dim. (hidden dimension), depth, head dim. (head dimension), and num. head (number of heads) denote the channel, the depth, and the dimension and the number of heads used in the attention layer used in the backbone TimeSformer.

| Dataset | x channel $C$ | Height $H'$ | Width $W'$ | Input patch size $(H/H', W/W')$ | Hidden dim. $C'$ | z channel $D$ | Depth | Head dim. | Num. head |
|---|---|---|---|---|---|---|---|---|---|
| UCF-101 | 4 | 32 | 32 | (2, 2) | 384 | 8 | 12 | 64 | 8 |
| WebVid-10M (SD1.5) | 4 | 32 | 32 | (2, 2) | 384 | 8 | 12 | 64 | 8 |
| WebVid-10M (SD2.1) | 4 | 32 | 64 | (2, 2) | 384 | 8 | 12 | 64 | 8 |

**Motion diffusion model.** We use the Adam optimizer with a learning rate of 1e-4, $(\beta_1, \beta_2) = (0.9, 0.999)$ and without weight decay. We use 8 and 32 NVIDIA A100 80GB GPUs to train the model on UCF-101 and WebVid (respectively) and use a batch size of 256. It takes 3-4 days for the convergence with 1024×512 resolution videos. Since the dimension of **c** of the text encoder and the hidden dimension in the motion diffusion model are different, we add and train a linear projection layer that maps the text hidden feature to the vector with the same dimension as the DiT hidden dimension. Our motion diffusion model implementation heavily follows the official implementation of DiT (Peebles & Xie, 2023), including hyperparameters and training objectives used.[*] We provide other hyperparameters related to the model configurations in Table 6.

Table 6: Hyperparameters related to our motion diffusion model.

| Dataset | Config. | Key. patch size | Input patch size | Text Encoder | Epochs | Ema |
|---|---|---|---|---|---|---|
| UCF-101 | DiT-L/2 | 4 | 2 | - | 3000 | 0.999 |
| WebVid-10M (SD1.5) | DiT-XL/2 | 4 | 2 | OpenClip (ViT/H) | 3 | 0.999 |
| WebVid-10M (SD2.1) | DiT-XL/2 | 8 | 2 | OpenClip (ViT/H) | 3 | 0.999 |

**Content frame diffusion model.** We use the Adam optimizer with a learning rate of 1e-4, $(\beta_1, \beta_2) = (0.9, 0.999)$ and without weight decay. We use 16 and 64 NVIDIA A100 80GB GPUs to train the model on UCF-101 and WebVid (respectively) and use a batch size of 256. For UCF-101 experiments, we train the model from scratch for a fair comparison with baselines. For WebVid-10M experiments, we fine-tune Stable Diffusion 1.5 and 2.1-base (Rombach et al., 2022) for zero-shot evaluation on UCF-101 and other evaluations (respectively), following the recent text-to-video generation works that use pretrained image models (Wang et al., 2023a). It takes 3-4 days for the convergence with 1024×512 resolution video frames due to high dimensionality of video frames. We provide other hyperparameters related to the model configurations in Table 7.

Table 7: Hyperparameters related to our content frame diffusion model.

| Dataset | Model | Epochs | Ema |
|---|---|---|---|
| UCF-101 (non-zero-shot) | DiT-XL/2 | 3000 | 0.999 |
| WebVid-10M (T2V) | SD 2.1 (base) | 3 | 0.999 |

### B.4 METRICS

**Sampler.** For both motion diffusion models and content frame diffusion models, we use the DDIM (Song et al., 2021a) sampler. We use $\eta = 0.0$ for both models (*i.e.*, without additional random noises in sampling), and we use the number of steps as 100 and 50 for the motion diffusion model and the content frame diffusion model, respectively. For the content frame diffusion model, we use the classifier guidance scale $w = 4.0$ on UCF-101 and $w = 7.5$ on text-to-video generation.

---

[*] https://github.com/facebookresearch/DiT

**CLIPSIM.** Following the protocol used in most text-to-video generation work (Singer et al., 2023; Wang et al., 2023b), we calculate CLIP scores (Wu et al., 2021) between a text prompt and generated video frames and report the average between them. Specifically, we each video frame into an image of resolution 224×224 and use it as an input to the CLIP image encoder. Following VideoLDM (Blattmann et al., 2023), we use the ViT-B/32 CLIP model (Radford et al., 2021).

**FVD.** For Fréchet video distance (FVD; Unterthiner et al. 2018), we mainly follow the recently fixed evaluation protocol proposed by StyleGAN-V (Skorokhodov et al., 2022). Specifically, this protocol samples a single random video clip from each video and extracts the feature using a pretrained I3D network (Carreira & Zisserman, 2017). For UCF-101 (non-zero-shot), we consider representative scenarios for evaluation: 2,048 real/fake samples (used in most previous methods such as DIGAN (Yu et al., 2022)) and 10,000 fake samples and clips in the training set (used in recent large-scale video generation methods such as MAGVIT (Yu et al., 2023a)). For zero-shot evaluation on UCF-101, we use the common protocol to generate 100 videos per class and calculate the fake statistics (Singer et al., 2023). For text prompts, we use the same text prompt used in PYoCo (Ge et al., 2023), as shown in Figure 8. For non-zero-shot evaluation on WebVid-10M, we use 5,000 video clips and text prompts in the validation set to calculate real and fake statistics, respectively. For FPS, we follow the exact same setup in the concurrent VideoLDM (Blattmann et al., 2023).

'applying eye makeup', 'applying lipstick', 'archery', 'baby crawling', 'gymnast performing on a balance beam', 'band marching', 'baseball pitcher throwing baseball', 'a basketball player shooting basketball', 'dunking basketball in a basketball match', 'bench press', 'biking', 'billiards', 'blow dry hair', 'blowing candles', 'body weight squats', 'a person bowling on bowling alley', 'boxing punching bag', 'boxing speed bag', 'swimmer doing breast stroke', 'brushing teeth', 'weightlifting with barbell', 'clean and jerk', 'cliff diving', 'bowling in cricket gameplay', 'batting in cricket gameplay', 'cutting in kitchen', 'diver diving into a swimming pool from a springboard', 'drumming', 'two fencers have fencing match indoors', 'field hockey match', 'gymnast performing on the floor', 'group of people playing frisbee on the playground', 'swimmer doing front crawl', 'golfer swings and strikes the ball', 'haircuting', 'a person hammering a nail', 'an athlete performing the hammer throw', 'an athlete doing handstand push up', 'an athlete doing handstand walking', 'massagist doing head massage to man', 'an athlete doing high jump', 'horse race', 'group of people racing horse', 'person riding a horse', 'a woman doing hula hoop', 'man and woman dancing on the ice', 'ice dancing', 'athlete practicing javelin throw', 'a person juggling with balls', 'a young person doing jumping jacks', 'a person skipping with jump rope', 'a person kayaking in rapid water', 'knitting', 'an athlete doing long jump', 'a person doing lunges with barbell', 'military parade', 'mixing in the kitchen', 'mopping floor', 'a person practicing nunchuck', 'gymnast performing on parallel bars', 'a person tossing pizza dough', 'a musician playing the cello in a room', 'a musician playing the daf', 'a musician playing the indian dhol', 'a musician playing the flute', 'a musician playing the guitar', 'a musician playing the piano', 'a musician playing the sitar', 'a musician playing the tabla', 'a musician playing the violin', 'an athlete jumps over the bar', 'gymnast performing pommel horse exercise', 'a person doing pull ups on bar', 'boxing match', 'push ups', 'group of people rafting on fast moving river', 'rock climbing indoor', 'rope climbing', 'several people rowing a boat on the river', 'couple salsa dancing', 'young man shaving beard with razor', 'an athlete practicing shot put throw', 'a teenager skateboarding', 'skier skiing down', 'jet ski on the water', 'sky diving', 'soccer player juggling football', 'soccer player doing penalty kick in a soccer match', 'gymnast performing on still rings', 'sumo wrestling', 'surfing', 'kids swing at the park', 'a person playing table tennis', 'a person doing TaiChi', 'a person playing tennis', 'an athlete practicing discus throw', 'trampoline jumping', 'typing on computer keyboard', 'a gymnast performing on the uneven bars', 'people playing volleyball', 'walking with dog', 'a person standing', 'doing pushups on the wall', 'a person writing on the blackboard', 'a kid playing Yo-Yo'

Figure 8: Text prompts used for zero-shot evaluation on UCF-101.

## C    MORE QUALITATIVE RESULTS

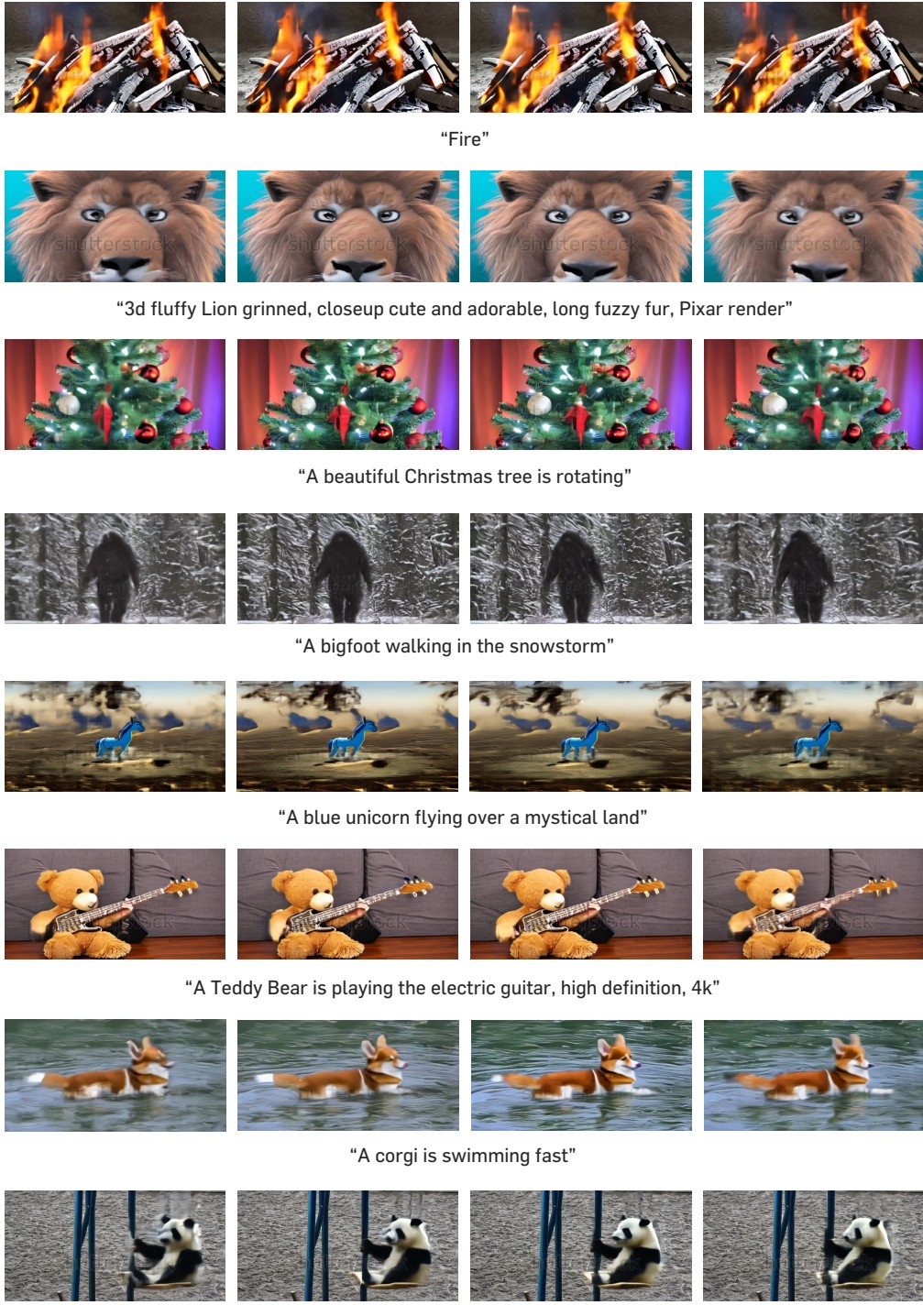

Figure 9: **512×1024 resolution, 16-frame text-to-video generation results** from our CMD.

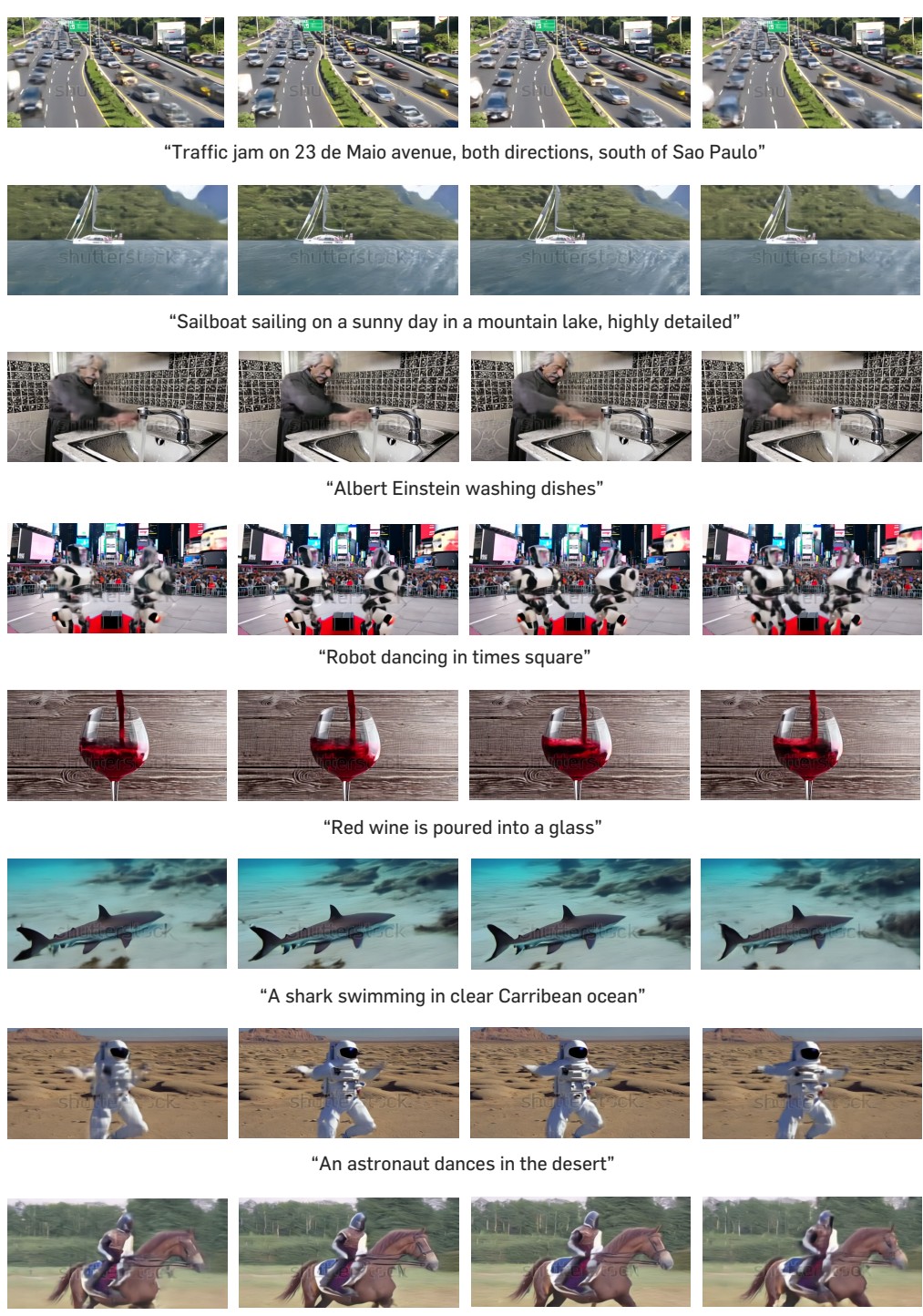

"Traffic jam on 23 de Maio avenue, both directions, south of Sao Paulo"

"Sailboat sailing on a sunny day in a mountain lake, highly detailed"

"Albert Einstein washing dishes"

"Robot dancing in times square"

"Red wine is poured into a glass"

"A shark swimming in clear Carribean ocean"

"An astronaut dances in the desert"

"A knight riding on a horse through the countryside"

Figure 10: **512×1024 resolution, 16-frame text-to-video generation results** from our CMD.

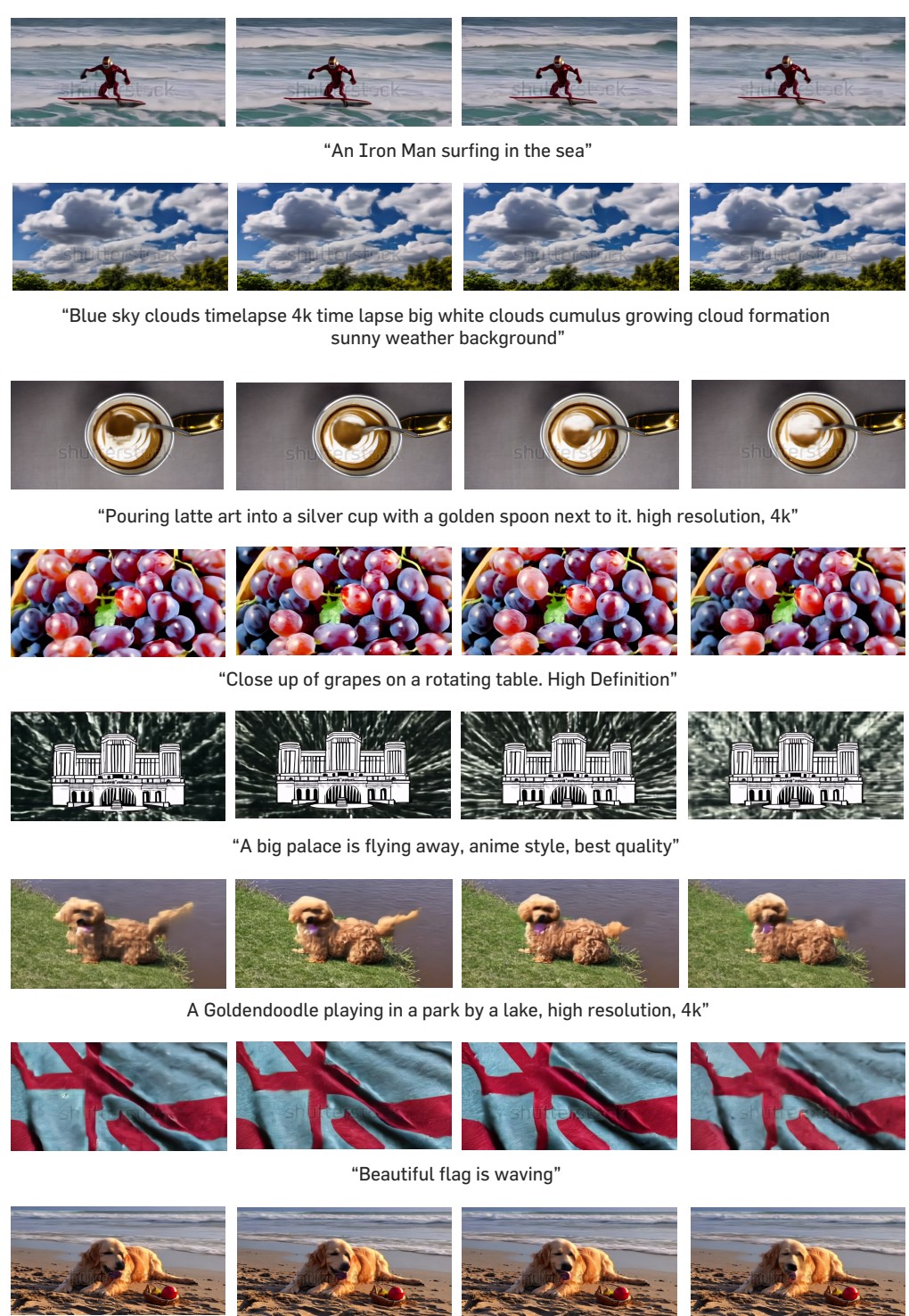

"An Iron Man surfing in the sea"

"Blue sky clouds timelapse 4k time lapse big white clouds cumulus growing cloud formation sunny weather background"

"Pouring latte art into a silver cup with a golden spoon next to it. high resolution, 4k"

"Close up of grapes on a rotating table. High Definition"

"A big palace is flying away, anime style, best quality"

A Goldendoodle playing in a park by a lake, high resolution, 4k"

"Beautiful flag is waving"

"A Golden Retriever has a picnic on a beautiful tropical beach at sunset, high resolution"

Figure 11: **512×1024 resolution, 16-frame text-to-video generation results** from our CMD.

# D  MORE DETAILS IN EFFICIENCY ANALYSIS

**Evaluation details.** For a fair comparison with baselines, we mainly use the official implementation provided in LVDM (He et al., 2022)[*] and ModelScope (Wang et al., 2023a).[*] For LVDM, we use `text2video.yaml` provided by the official implementation that uses a spatial downsample factor of 8 and a temporal downsample factor of 1. We adjust all factors and techniques that affect memory and computation dramatically. First, all values are measured with the same machine with a single NVIDIA A100 40GB/80GB GPU. Moreover, we use mixed precision operation and memory-efficient attention mechanisms in all baselines.[*] We also use half precision (fp16) for measuring the efficiency in sampling. Finally, we disable gradient checkpointing for all baselines in measuring the training efficiency. For measuring floating point operations (FLOPs), we use the Fvcore library that measures the FLOPs of Pytorch models.[*]

**Exact values.** In Table 8 and 9, we provide exact values of each bar in Figure 5 and 6.

Table 8: Exact values of component-wise efficiency analysis.

| Method | Model | TFLOPs | sec/step | Memory (GB) |
|---|---|---|---|---|
| ModelScope | Diffusion | 9.41 | 0.920 | 69.6 |
| LVDM | Diffusion | 6.28 | 0.456 | 44.2 |
| | Autoencoder | 0.77 | 0.109 | 8.93 |
| CMD | Content Frame Diffusion | 0.34 | 0.151 | 21.5 |
| | Motion Diffusion | 0.14 | 0.100 | 18.7 |

Table 9: Exact values of sampling efficiency analysis.

| Method | TFLOPs | | Time (s) | | Memory (GB) | |
|---|---|---|---|---|---|---|
| | Step=50 | Step=100 | Step=50 | Step=100 | Batch=1 | Batch=4 |
| ModelScope | 939.0 | 1877.9 | 35.1 | 70.3 | 8.50 | 21.3 |
| LVDM | 625.6 | 1251.2 | 24.2 | 48.1 | 8.38 | 18.6 |
| CMD | 46.8 | 92.1 | 3.13 | 6.05 | 5.56 | 8.57 |

**Model size comparison.** In Table 10, we summarize the number of model parameters of recent text-to-video generation models. Our model has a similar size to ModelScope (Wang et al., 2023a) and is much smaller than popular, well-performing text-to-video generation models, *e.g.*, VideoLDM and Imagen-Video. In this respect, we strongly believe the generation quality of our CMD will be much stronger if one enlarges the overall model sizes by adjusting model configurations.

Table 10: Model size analysis.

| Method | ModelScope | LVDM | VideoLDM | Imagen-Video | PyoCo | CogVideo | CMD (Ours) |
|---|---|---|---|---|---|---|---|
| # params. | 1.7B | 0.96B | 3.1B | 11.6B | 2.6B | 9B | 1.6B |

---

[*] https://github.com/YingqingHe/LVDM
[*] https://huggingface.co/spaces/damo-vilab/modelscope-text-to-video-synthesis
[*] https://github.com/facebookresearch/xformers
[*] https://github.com/facebookresearch/fvcore

# E  SAMPLING PROCEDURE

We summarize the sampling procedure of CMD in Algorithm 1.

---

**Algorithm 1** content-motion latent diffusion model (CMD)

---

1: Sample random Gaussian noise $\bar{\mathbf{x}}_T \sim \mathcal{N}(\mathbf{0}_{\bar{\mathbf{x}}}, \mathbf{I}_{\bar{\mathbf{x}}})$, $\mathbf{z}_T \sim \mathcal{N}(\mathbf{0}_{\mathbf{z}}, \mathbf{I}_{\mathbf{z}})$.
2: **for** $t = T$ to 1 **do**
3:     $\boldsymbol{\epsilon}_t = (1 + w)\boldsymbol{\epsilon}_{\boldsymbol{\theta}_I}(\bar{\mathbf{x}}_t, \mathbf{c}, t) - w\boldsymbol{\epsilon}_{\boldsymbol{\theta}_I}(\bar{\mathbf{x}}_t, \mathbf{0}, t)$.
4:     Apply a pre-defined sampler (*e.g.*, DDIM (Song et al., 2021a)) to $\bar{\mathbf{x}}_{t-1}$ from $\bar{\mathbf{x}}_t$ and $\boldsymbol{\epsilon}_t$.
5: **end for**
6: **for** $t = T$ to 1 **do**
7:     $\boldsymbol{\epsilon}_t = (1 + w)\boldsymbol{\epsilon}_{\boldsymbol{\theta}_M}(\mathbf{z}_t, \mathbf{c}, \bar{\mathbf{x}}_0, t) - w\boldsymbol{\epsilon}_{\boldsymbol{\theta}_M}(\mathbf{z}_t, \mathbf{0}, \bar{\mathbf{x}}_0, t)$.
8:     Apply a pre-defined (*e.g.*, DDIM (Song et al., 2021a)) to $\mathbf{z}_{t-1}$ from $\mathbf{z}_t$ and $\boldsymbol{\epsilon}_t$.
9: **end for**
10: Decode the clip from latents: $\mathbf{x}^{1:L} = g_{\boldsymbol{\theta}}(\bar{\mathbf{x}}_0, \mathbf{z}_0)$.
11: Output the generated video $\mathbf{x}^{1:L}$.

---

## F  ROLE OF MOTION LATENT VECTORS

As shown in Figure 12, given a fixed content frame, our method can generate videos with different motions, by generating different motion latent vectors.

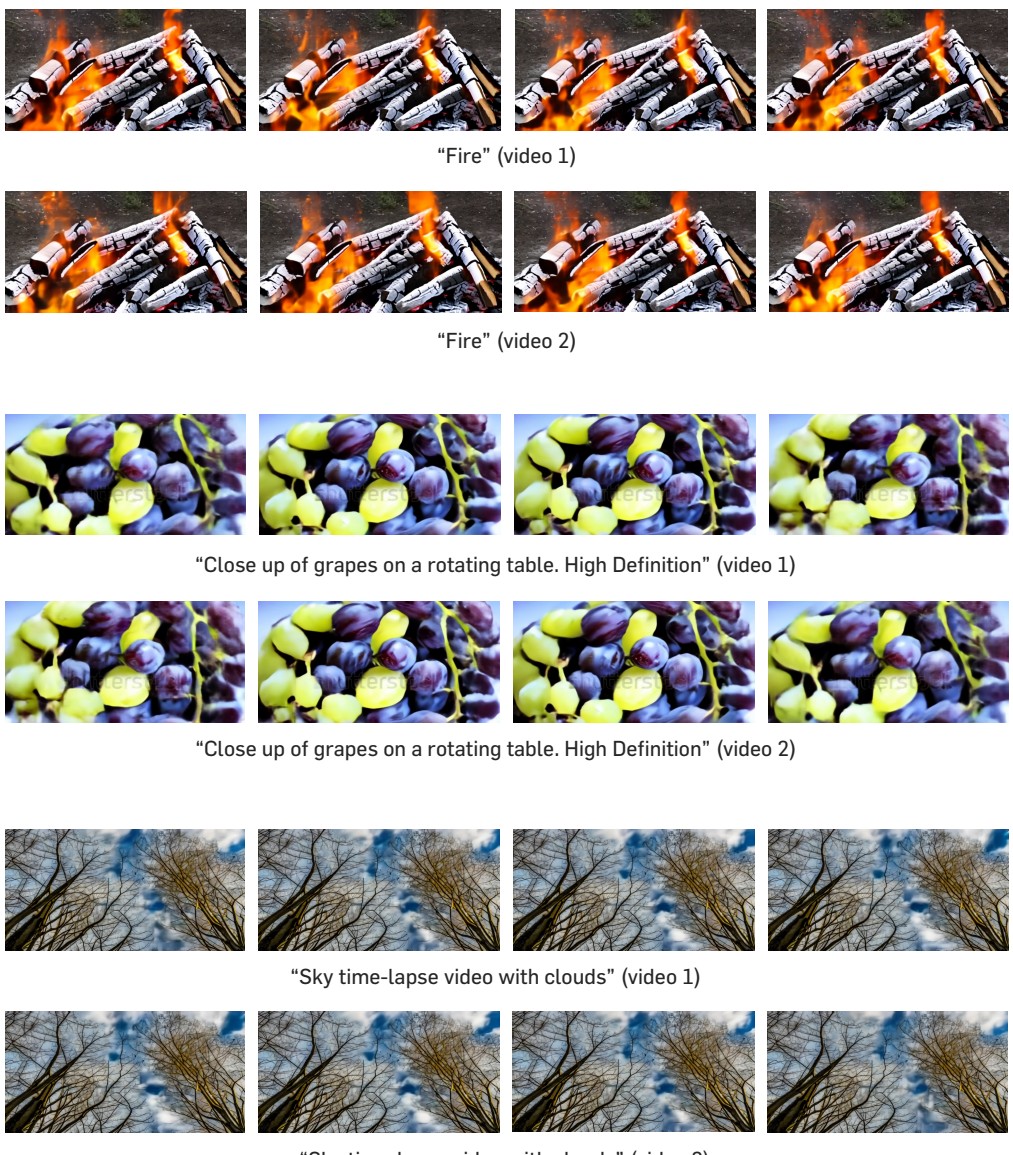

Figure 12: **512×1024 resolution, 16-frame text-to-video generation results** from CMD, where we fix a content frame for each text prompt and sample different motion latent vectors.

# G QUANTITATIVE RESULTS WITH DIFFERENT GUIDANCE SCALES

In Table 11, we report FVD and CLIPSIM scores with different classifier-free guidance (cfg) scales.

Table 11: **T2V generation on WebVid-10M.** ↓ and ↑ indicate lower and higher scores are better, respectively.cfg denotes classifier-free guidance scale.

| Method | FVD ↓ | CLIPSIM ↑ |
|---|---|---|
| CMD (cfg=7.0) | 235.5 | 0.3001 |
| CMD (cfg=9.0) | 238.3 | 0.3020 |
| CMD (cfg=10.0) | 245.2 | 0.3031 |
| CMD (cfg=11.0) | 246.9 | 0.3034 |

## H    QUALITATIVE RESULTS OF THE AUTOENCODER

In Figure 13 and 14, re visualize ground truth frames in the validation set in the WebVid-10M dataset (left) and reconstructions from our autoencoder (right). To visualize frames better, we only present the first frame in the paper; please refer to the supplementary material for video visualization.

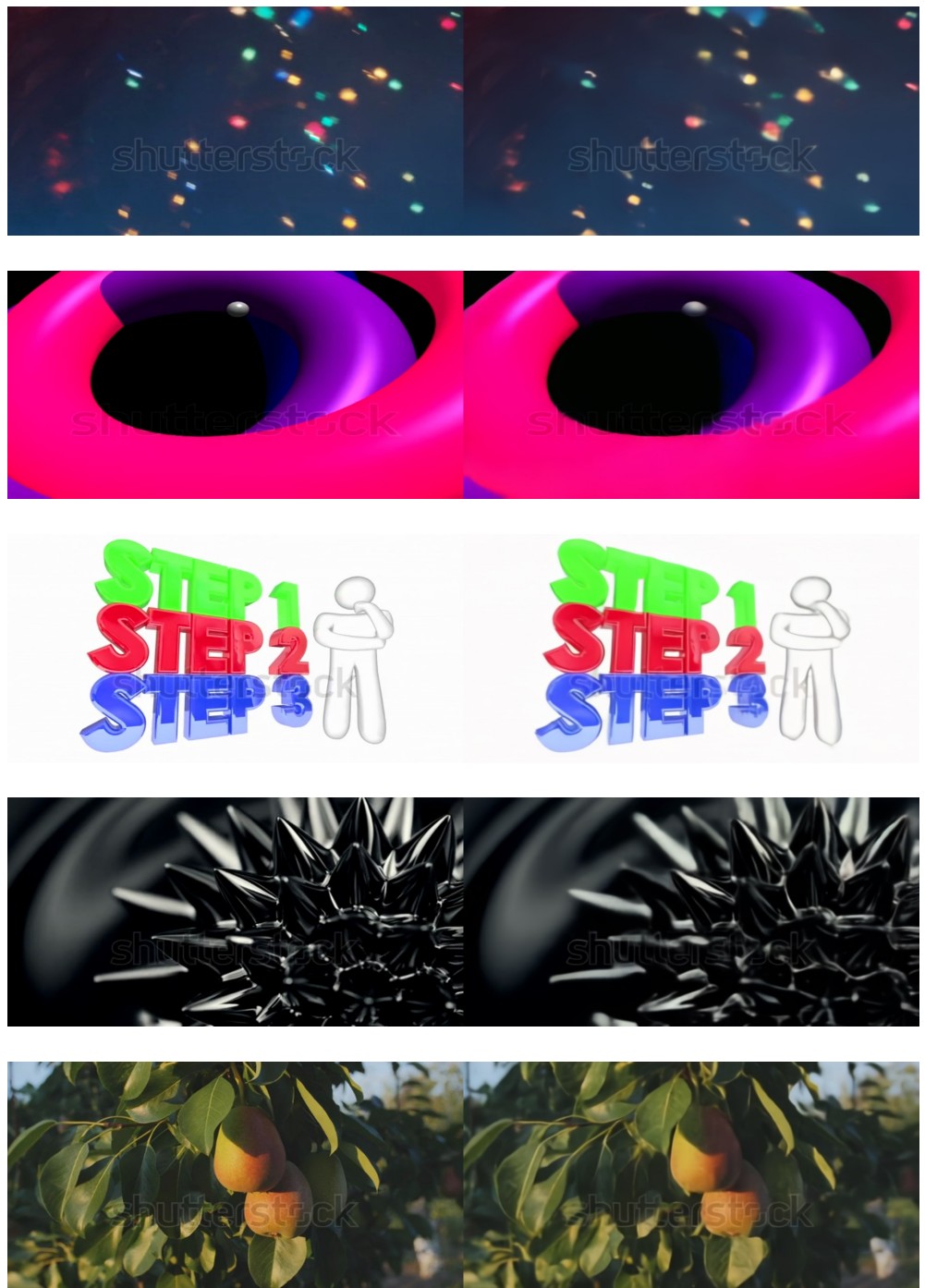

Figure 13: **512×1024 resolution, 16-frame video reconstruction results** from our CMD.

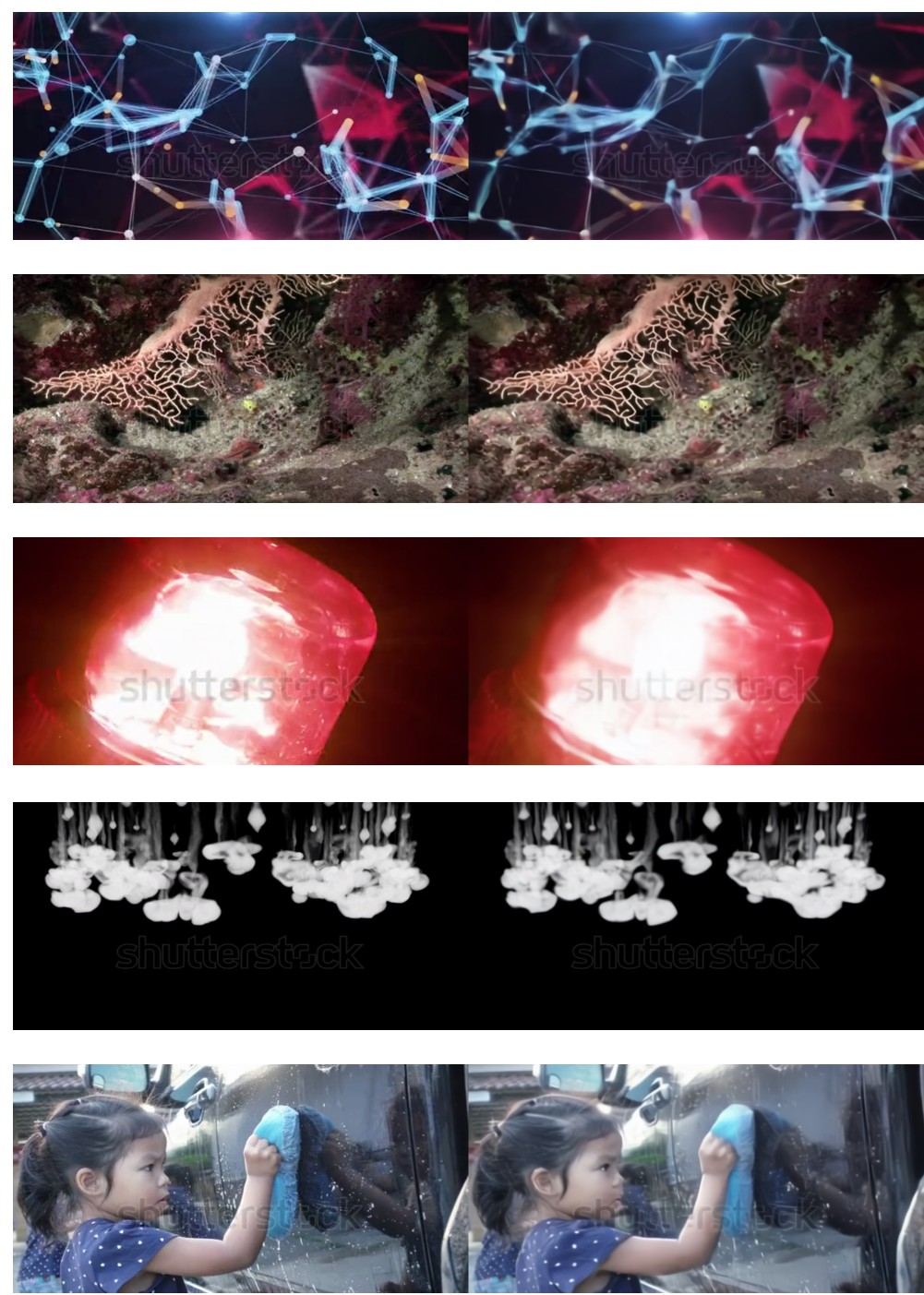

Figure 14: **512×1024 resolution, 16-frame video reconstruction results** from our CMD.

# I LIMITATION AND FUTURE WORKS

While CMD shows promising results on existing video generation benchmarks, there exist several limitations and corresponding future works. In what follows, we explain such limitations in addition to the limitations that we mentioned in our main text.

**Autoencoder quality.** We found that our keyframe design works very well in pixel space, as also shown in promising results for the UCF-101 generation. While this concept also fairly worked well in latent space built in an image-wise manner (*e.g.*, Stable Diffusion latent space (Rombach et al., 2022)), we found there exists considerable frame-wise quality drop if the underlying motion in the video contains extremely dynamic motion. We hypothesize this is because the latent space that we used for reconstruction does not consider the temporal coherency of videos, resulting in less temporally coherent frames as latent vectors than frames in pixel space. We believe this limitation can be mitigated by the following solutions. First, future works can consider training our module in low-resolution pixel space first with training additional upsampler diffusion models (*i.e.*, cascaded diffusion). Moreover, one can consider constructing the latent space from scratch using both large image and video data. Finally, our keyframe design as a weighted sum of video frames may not be an optimal choice to represent the overall contents of the video; exploring the better forms of content vectors that are similar to the original image should be an interesting future work.

**Model size.** We use fairly small autoencoder and diffusion models due to lack of resources used for training, compared with recent text-to-video generation models. Exploring the quality improvement with respect to the number of model parameters also should an interesting direction.

**Using negative prompts.** We do not apply negative prompts in text-to-video generation, which have been recently used to improve the generated video quality. We believe that applying this technique to CMD will improve the video quality.

