# OpenReview forum: "Efficient Video Diffusion Models via Content-Frame Motion-Latent Decomposition"
_ICLR.cc/2024/Conference — ICLR 2024 poster_

### Official Review · Reviewer_BXkg · 2023-10-30

**Soundness:** 2 fair
**Presentation:** 3 good
**Contribution:** 2 fair
**Rating:** 6
**Confidence:** 5

**Summary:**

This paper proposes a video diffusion model that can be trained efficiently via content-motion disentanglement. An autoencoder is trained to learn common visual content and low-dimensional video motion. Then, a pretrained image diffusion model is fine-tuned to generate content frames, and the motion patents are generated with a lightweight diffusion model. This method can generate a video faster while achieving better FVD in WebVid-10M.

**Strengths:**

1.	CDM consumes much lower computation resources to inference.
2.	The method presented in this paper is cool and has the potential to improve the video generation research community, especially the idea of decomposing a video to motion and content.
3.	CDM achieves good CLIPSIM metric in MSR-VTT and WebVid-10M.
4.	The GPU memory usage and speed evaluations are extensive.

**Weaknesses:**

This is a technically solid paper. I appreciate the design of the autoencoder design in CDM and believe the video generation community can learn from it. But it suffers from the following weaknesses:
1.	Experiment comparison setting: According to the supplementary video, CDM’s generation has considerably slow motion or low motion amplitude. However, LVDM and ModelScope can produce diverse and large motions. Therefore, the FVD comparison setting is not fair. We need other experiments to show CDM is better. For example, LVDM is trained on WebVid-10M with frame stride 8 (i.e. sample 16 keyframes in a 128-frame video clip), please re-train CDM to match the setting of LVDM.
2.	Evaluation of autoencoder: Please conduct a quantitative evaluation of the performance of the CDM autoencoder. You can use a series of quantitative metrics. For example, SSIM, PSNR, etc. It would be great if you also showed how good stable-diffusion KLVAE can do on video reconstruction.
3.	Training cost: To train CDM, one needs to train an autoencoder, a content diffusion, and a motion diffusion model. As a result, the training cost of CDM is very close to that of training ModelScope or LVDM. According to the supplementary material, training the autoencoder consumes 8 A100 for a week. After that, content diffusion and motion diffusion models can be trained. And they use 96 A100 for days to train on WebVid-10M. The training cost of CDM is not considerably lower than LVDM and ModelScope.
4.	Visual definition problem: CDM produces 512x1024 resolution videos. However, the supplementary is a 9M mp4 file, and is very hard to tell the visual definition of the samples. It would be great if the author provided evaluations on the visual definition of CDM.

This paper is generally well-written. But when I parse through the literature, I find some statements that are somewhat unproper. See below:
1.	In the abstract, the author says CDM is the first that can directly utilize a pretrained image diffusion model. But, to the best of my knowledge, Video LDM, LVDM, and ModelScope are also utilizing pretrained text-to-image models.
2.	Please number your equations. It will be very helpful for readers to understand your math formulas.
3.	The training time for the content frame diffusion generator is not shown in the supplementary material.

**Questions:**

Please see the above questions.

---

> ### Author Response · Authors · 2023-11-21
> **Response to Reviewer BXkg**
>
> We deeply appreciate your insightful comments and efforts in reviewing our manuscript. We mark our major revision in the revised manuscript with “blue.” We respond to each of your comments one-by-one in what follows.
>
> ---
>
> **[W1] Experiment comparison setting: Adjust frame sampling stride with LVDM.**
>
> As mentioned in Appendix B, we followed the setup of VideoLDM [1] to show our method can generate videos with smooth motions. Nevertheless, we agree that adjusting FPS with LVDM and ModelScope will provide a more fair comparison, and we re-trained CMD with an adjusted frame sampling rate that you suggested and reported the FVD score in the Table below. As shown in this table, our method still shows a better FVD than baselines.
>
> | Method       | FVD   |
> |--------------|-------|
> | LVDM         | 455.5 |
> | ModelScope   | 414.1 |
> | VideoFactory | 292.4 |
> | CMD (Ours)   | **238.3** |
>
> [1] Align your Latents: High-Resolution Video Synthesis with Latent Diffusion Models, CVPR 2023
>
> ---
>
> **[W2] Quantitative evaluation of autoencoder.**
>
> Thanks for your constructive comments. Besides the quantitative evaluation of our autoencoder using reconstruction FVD in Table 4(b), we additionally evaluate our autoencoder with the metrics you suggested, SSIM and PSNR, and report them in the Table below. While our autoencoder (CMD) has a 9.14$\times$ smaller dimension in the latent space of encoding videos, it still shows reasonable reconstruction quality.
>
> | Method           | Dimension | R-FVD | SSIM  | PSNR  |
> |------------------|-----------|-------|-------|-------|
> | CMD              | 57344     | 56.80 | 0.863 | 28.71 |
> | Stable diffusion | 524288    | 24.26 | 0.914 | 33.58 |
>
> ---
>
> **[W3] Training cost in Appendix seems not considerably lower than LVDM and ModelScope.**
>
> There are two factors that make a direct comparison in terms of the total training GPU hours difficult. First, to our best knowledge, we could not find the exact total GPU hours for training LVDM and ModelScope. If the reviewer can provide a reference on that, we will be happy to perform a comparison and add the comparison and incorporate that in the manuscript. Second, the training cost of CMD in Appendix is measured with video clips of resolution 512$\times$1024 with a length 16, which is significantly higher-dimensional than the data preprocessed in LVDM and ModelScope (which is 256$\times$256 with a length 16). Hence, if the reported training cost of CMD is similar to (or slightly lower than) LVDM or ModelScope, it implies CMD can be trained much more efficiently than baselines.
>
> Given the above reasons, we instead measured the training cost of our method and baselines in the same normalized setup (i.e., batch size 1 and a single forward pass with the same resolution). As shown in Figure 5, CMD has much lower training cost than other baselines in our evaluation setting.
>
> ---
>
> **[W4] Visual definition problem: Better high-resolution visualization?**
>
> To address your concern, we visualized each video with a single text prompt and saved them to different video files (in total 28 videos). We attached them to the supplementary material.
>
> ---
>
> **[C1] There are other video diffusion models that utilizes pretrained text-to-image diffusion models.**
>
> As you mentioned, several works use pretrained text-to-image for text-to-video generation. However, they need to add some temporal layers inside the text-to-image models to deal with a sequence of image-wise latent vectors, which is both memory and computation inefficient. Our approach is different: CMD directly fine-tunes the text-to-image model without adding any additional parameters inside the text-to-image model, enabling efficient training and inference. Nevertheless, we also agree that the current abstract might mislead readers and thus we revised it accordingly.
>
> ---
>
> **[C2] Add a number to equations.**
>
> Thanks for your constructive comments. We now added equation numbers in the revised manuscript.
>
> ---
>
> **[C3] Training time for the content frame diffusion generation is now shown.**
>
> Thank you for pointing this out. We now added training time for content frame diffusion in Appendix B.3.

---

### Official Review · Reviewer_cm4G · 2023-10-31

**Soundness:** 3 good
**Presentation:** 3 good
**Contribution:** 3 good
**Rating:** 8
**Confidence:** 4

**Summary:**

This paper presents the content-motion latent diffusion model (CMD) for video generation. The CMD model consists of an autoencoder and two diffusion models learned in two stages. In the first stage, the autoencoder is trained to map an input video into a lower-dimensional latent representation. This latent representation consists of a single “content frame” (a weighted sum of the input frames) and a latent vector encoding motion information. In the second stage, CMD fine-tunes a pre-trained image diffusion model to learn the distribution of the content frames and additionally trains another diffusion model to generate the latent motion representation conditioned on a given content frame. CMD is compared to several baselines on popular video generation benchmarks such as UCF-101 and WebVid-10M and achieves a comparable or better performance (measured by metrics such as FVD and CLIPSIM) while maintaining lower memory and computational costs.

**Strengths:**

- The proposed CMD model achieves a comparable or better performance (measured by metrics such as FVD and CLIPSIM) while maintaining lower memory and computational costs against several baselines.

- CMD can utilize a pre-trained image diffusion model which can save training costs.

- With minor exceptions (addressed in the questions section), the paper is written clearly.

- To the best of my knowledge, the CMD model is novel and would be of interest to the ICLR community.

**Weaknesses:**

- Since the focus is on relatively short videos (16 frames), it would be interesting to establish whether CMD’s downstream performance is still favorable as the number of frames increases.

- The presented results can be even stronger if confidence intervals are provided.

**Questions:**

- Would it be feasible to include confidence intervals for results in Tables 1-3?

- Could the authors provide more intuition behind the 2D-projection-based encoding for the motion latent?

- In Figure 5, wouldn’t the comparison between different models’ computational & memory consumption be represented more accurately if there is a single bar for the CMD model with three stacked parts (autoencoder, content frame DM, motion latent DM)? For example, in the Memory panel, it would show that CMD has higher memory consumption than LVDM.

- Additionally, how were the baselines (ModelScope, LVDM) chosen in Figure 5?

- Are the authors planning to release an implementation of their proposed framework?

Typos:
- Abstract: “can directly utilizes a pretrained” -> “can directly utilize a pretrained”.

---

> ### Author Response · Authors · 2023-11-21
> **Response to Reviewer cm4G**
>
> We deeply appreciate your insightful comments and efforts in reviewing our manuscript. We mark our major revision in the revised manuscript with “blue.” We respond to each of your comments one-by-one in what follows.
>
> ---
>
> **[W1] Performance of CMD when the number of frames increases?**
>
> We fully agree that it would be an exciting future direction to extend our framework for longer video generation, and we strongly believe CMD has a strong potential for such a goal. First, we provided the performance of an autoencoder with a longer length of 24 in Table 4. It still shows a reasonable reconstruction performance, and thus, we think our CMD can be extended for longer video clips. For extremely long videos (e.g., a length of thousands), our method can be extended as well by following the strategy in PVDM [1] that divides a long video into multiple short video clips, encodes each video clip into succinct latents, and then perform clip-by-clip generation conditioned on a previous clip.
>
> [1] Yu et al., Video Probabilistic Diffusion Models in Projected Latent Space, CVPR 2023.
>
> ---
>
> **[W2, Q1] Confidence interval of the method.**
>
> Thank you for your constructive comments. We added a standard deviation of CMD in Table 1 in the revised manuscript. We also added a standard deviation of baselines if it is reported from references. For Table 2 and 3, it is difficult to add confidence intervals of baselines, because most baselines have not reported them in their quantitative evaluation. Nevertheless, we fully agree that at least providing confidence intervals of CMD in Table 2 and 3 will help future readers and research, and we will add them to the final manuscript. Please understand it was difficult to add in the revision due to the limited resources we have.
>
> ---
>
> **[Q2] More intuition behind 2D-projection-based encoding for the motion latents.**
>
> Our motivation is based on recent single-video encoding [2] and video dataset encoding [1] via 2D projections. This is possible as videos are temporally coherent 3D arrays with common contents (e.g., background or moving objects) and thus can be parameterized succinctly by optimizing 2D planes across each axis to capture them. Accordingly, we use triplane latent space to encode video datasets using projections along with a popular video encoder/decoder. This is also related to the recent 3D generation method [3] that encodes 3D voxels succinctly with 2D triplanes.
>
> [2] Kim et al., Scalable Neural Video Representations with Learnable Positional Features, NeuriPS 2022.
> [3] Chan et al., EG3D: Efficient 3D Geometry-aware Generative Adversarial Networks, CVPR 2022.
>
> ---
>
>
> **[Q3] Single bar for the CMD with three stacked parts for a fair efficiency comparison (Figure 5)?**
>
> We did not report the stacked values for Figure 5, since training of three components do not need to be done simultaneously. Specifically, we do not need to allocate memory for three models at once since each model can be trained separately, so we just provided the component-wise memory and computation consumption.
>
> Nevertheless, we agree that for computation efficiency (FLOPs and sec/step) in training, stacking all components of CMD can help readers understand the computation efficiency of entire pipelines in handling a single video batch. Our method has 1.25 TFLOPs and 0.360 sec/step, which is still much better than 6.28 TFLOPs and 0.456 sec/step of LVDM.
>
> ---
>
> **[Q4] How were the baselines chosen in Figure 5?**
>
> As mentioned in Section 4.3, we chose text-to-video diffusion models as baselines if their implementation and checkpoints are publicly available.
>
> ---
>
> **[Q5] Plan for releasing an implementation.**
>
> We are planning to release the implementation.
>
> ---
>
> **[C1] Editorial comment.**
>
> Thanks for your careful reading. We fixed the typo in the revised manuscript.

---

> > ### Comment · Reviewer_cm4G · 2023-11-22
> > **Thank you for your reponse**
> >
> > Thank you for your insightful clarifications and the addition of confidence intervals for some of the results in Table 1. I do not have any follow-up questions at the moment.

---

> > > ### Author Response · Authors · 2023-11-23
> > > **Thank you for your comments!**
> > >
> > > We are happy to hear that our response could help to address your concerns!
> > >
> > > Due to your valuable and constructive comments and suggestions, we do believe that our work is much more improved.
> > > If you have any further questions or concerns, please do not hesitate to let us know.
> > >
> > > Thank you very much,
> > > Authors

---

### Official Review · Reviewer_jHjr · 2023-11-01

**Soundness:** 3 good
**Presentation:** 3 good
**Contribution:** 3 good
**Rating:** 8
**Confidence:** 4

**Summary:**

This paper proposes to decompose a  video clip into a content frame and a motion vector, both of which are then generated by two latent diffusion models accordingly. The proposed method largely accelerates text-to-video generation while maintaining high generation qualtiy.

**Strengths:**

1. The paper is well-written and easy to follow. The proposed idea is illustrated clearly.

2. The idea of training an auto-encoder to decompose the video into content & motion is novel and dose help accerlate the generation of videos.

3. The experiments are adequate to illustrate the superiority of the proposed method.

**Weaknesses:**

1. Constrained by computational resources, the auto-encoder can only handle a limited number of frames at one time, which means that the auto-encoder can only encode motions within a limited interval. The reduced computational cost does not help us to explore the generation of longer videos.

2. The content code is represented by the weighted sum of original frames, which may limit the representing ability of the content code. For example, the key components of a video may appear in different frames, (e.g. one person in frame 1 and another person in frame 2), in which case, it is difficult to maintain the information all key components in the content code.

**Questions:**

What is the resolutoin of the generated videos in Table 1, 2 and 3?

---

> ### Author Response · Authors · 2023-11-21
> **Response to Reviewer jHjr**
>
> We deeply appreciate your insightful comments and efforts in reviewing our manuscript. We mark our major revision in the revised manuscript with “blue.” We respond to each of your comments one-by-one in what follows.
>
> ---
>
> **[W1] Constrained by computational resources, the auto-encoder can only handle a limited number of frames at one time.**
>
> As you mentioned, there exists a limit length to an autoencoder that can encode videos as succinct latent vectors at once. Nevertheless, we believe that it does help in exploring generations of longer videos. Specifically, for longer videos, one can consider dividing it into multiple video clips of a fixed length, encode each clip as latent vectors using our autoencoder, and then consider modeling these multiple latent vectors. Such a strategy has also been widely adopted in latent video generative modeling literature [1,2]. While we did not try this due to the lack of resources, we strongly believe our method can be similarly extended for longer videos.
>
> [1] Villegas et al., Phenaki: Variable Length Video Generation From Open Domain Textual Description, ICLR 2022.
> [2] Yu et al., Video Probabilistic Diffusion Models in Projected Latent Space, CVPR 2023.
>
> ---
>
> **[W2] Content code as a weight sum of the original frame may limit the representing ability.**
>
> As you stated, content frames alone as a weighted sum of the original frame might limit representation capability. However, at the same time, this also constrains the space of content code to easily exploit pretrained image diffusion models. Also, recall that we also included "motion latents" in the autoencoder design, which can compensate for the representing ability of content frames. Because of this, our encoding scheme with this content code design can encode video with a high quality even if the scene change is dramatic, as empirically shown in Table 4a and 4b. Exploring a better design of content code and motion latent vectors should be an interesting future work.
>
> ---
>
> **[Q1] What is the resolution of the generated videos in Table 1, 2, and 3?**
>
> All of the video frames in Figure 1 and 2 have 512$\times$1024 resolution.

---

### Official Review · Reviewer_cbVn · 2023-11-05

**Soundness:** 3 good
**Presentation:** 3 good
**Contribution:** 2 fair
**Rating:** 6
**Confidence:** 2

**Summary:**

The paper proposes to generate videos by decomposing the latent representation into content-related features and motion features. The proposed method consists of two blocks: decomposed autoencoder and diffusion models for motion sampling. The proposed method generates plausible videos.

**Strengths:**

1) The paper is well-written and easy to follow.

2) The proposed method achieves state-of-the-art performance in video generation.

3) It makes sense to decompose the video generation into two parts, the content generation and the motion part.

**Weaknesses:**

1) The idea of decomposing the video generation into two factors has been proposed in previous methods [a, b, c].

[a] Yuwei Guo, Ceyuan Yang, Anyi Rao, Yaohui Wang, Yu Qiao, Dahua Lin, Bo Dai. AnimateDiff: Animate Your Personalized Text-to-Image Diffusion Models without Specific Tuning

[b] Yuming Jiang, Shuai Yang, Tong Liang Koh, Wayne Wu, Chen Change Loy, Ziwei Liu. Text2Performer: Text-Driven Human Video Generation

[c] Haomiao Ni, Changhao Shi, Kai Li, Sharon X. Huang, Martin Renqiang Min. Conditional Image-to-Video Generation with Latent Flow Diffusion Models.

2) In the training of autoencoder, how to ensure the model learns meaningful motion representation, rather than directly copying the information containing content information from the inputs?

3) The content frame is an average of the latent representations of multiple frames, and this would lead to the generated frames being blurry, especially for the first frame. From the results of Fig. 10 and Fig. 11, the first frames are kindly of blurry.

**Questions:**

Please see my concerns in the weakness part.

---

> ### Author Response · Authors · 2023-11-21
> **Response to Reviewer cbVn**
>
> We deeply appreciate your insightful comments and efforts in reviewing our manuscript. We mark our major revision in the revised manuscript with “blue.” We respond to each of your comments one-by-one in what follows.
>
> ---
>
> **[W1] Idea of decomposing video generation into two factors has been proposed in previous methods**
>
> We agree that previous works have shared a similar idea of decomposing videos into two factors. However, the key difference between CMD and them is in the purpose of decomposition. In this work, we explored a new video decomposition that (a) exploits pretrained image diffusion models for video generation (b) in computation- and memory-efficient manner. To achieve this, we introduced an encoding scheme in which one of the factors (content frame) lies in the image space (for (a)), and the remaining part (motion latent vectors) becomes a low-dimensional vector, and thus we train an additional lightweight diffusion model to predict these motion latents vectors (for (b)).
>
> Unlike our method, existing approaches have not considered (a) and (b) simultaneously. Specifically, [1] exploits pretrained image diffusion models, but it deals with videos as high-dimensional cubic tensors directly. Moreover, [2, 3] do not use pretrained image diffusion models, and [3] uses flow for motion encoding, which is a high-dimensional cubic tensor as well. Nevertheless, we think adding a discussion with relevant work [1, 2, 3] that you mentioned will give better insight to future readers, and we added a related discussion in Appendix A in the revised manuscript.
>
> [1] Guo et al.,. AnimateDiff: Animate Your Personalized Text-to-Image Diffusion Models without Specific Tuning
> [2] Jiang et al., Text2Performer: Text-Driven Human Video Generation
> [3] Ni et al., Conditional Image-to-Video Generation with Latent Flow Diffusion Models.
>
>
>
> ---
>
> **[W2] How to ensure the model learns meaningful motion representation?**
>
> This is a good point. In our video autoencoder, the content frame is a static image without the time dimension, so without meaningful motion representation learned by the motion latent, it cannot reconstruct the video. In other words, it is the video reconstruction and content-motion decomposition in our video autoencoder that ensures the motion latent captures the meaningful motion representation. Empirically, we also found our motion latent vectors indeed contain meaningful motion information. For instance, when generating time-lapse videos of the sky, our model can generate videos with diverse cloud-moving directions at a given fixed content frame. We included this result in Appendix F in the revised manuscript.
>
> ---
>
> **[W3] Generated frames being blurry due to the design of the content frame**
>
> Thanks for pointing this out. In principle, since our design includes not only a content frame but also a “motion latent vector”, the learned motion latent will compensate for the representing power of the content frame, and reconstruct each frame with a high quality. This has been empirically supported by the high-quality encoding performance in Table 4a and 4b. In practice, we think the appearance of blurriness in generated videos may be attributed to a relatively not large dataset and a relatively small model size. Hence, we believe visual quality can be much better if we use a larger text-video dataset, as validated in the concurrent work [4]. Moreover, recall that our model includes a “single” diffusion model that has a relatively small model size (1.7B including the autoencoder) in total without any cascaded modules, unlike the existing video diffusion models; we also believe training larger models will greatly improve the frame quality.
>
> [4] Wang et al., VideoFactory: Swap Attention in Spatiotemporal Diffusions for Text-to-Video Generation, arXiv 2023

---

> > ### Comment · Reviewer_cbVn · 2023-11-23
> > **Post rebuttal**
> >
> > Dear authors,
> >
> > After reading the rebuttal, I still have some questions:
> >
> > [W2] I agree that the model needs both content frame and motion latent representation for reconstruction. However, my question is since you will feed the frames into the model to extract the motion latent representation, will the model directly encode content information as well as motion information into the motion latent representation? In this case, the model would not need a content frame to reconstruct the target frames. Do you add any constraints to avoid the model of learning this naive solution?
> >
> > [W3] I am not convinced by the explanation for the cause of the blurry frames. From Table 4a and 4b, from the quantitative results, we cannot tell if the reconstructed frames are blurry or not. Can you provide some visual results to support your hypothesis?
> >
> > Thanks,

---

> ### Author Response · Authors · 2023-11-23
> **Response to Reviewer cbVn**
>
> We are truly grateful for taking your time to provide additional recommendations and acknowledge our efforts.
>
> ---
>
> **[W2]** We use an extremely low-dimensional motion vector as the constraint. For instance, in our video experiments to encode 512$\times$1024 resolution RGB frames with a length of 16, the dimensionality of the motion latent vector is 24,576, 0.1% of the original dimensionality). Such a low dimensionality gives a strong bottleneck to encode the entire video without using content frames; hence, the autoencoder should utilize rich spatial information included in the content frame to achieve high-quality encoding. Also, in Appendix F, we have shown that fixing the content frame maintains the overall contents of the generated videos even if it is combined with different motion latent vectors. This result also empirically verifies that our autoencoder does use the information in the content frame, not just solely relying on the motion latent vector to encode the videos.
>
> ---
>
> **[W3]** To further address your concern, we provide some qualitative results in Appendix H and the supplementary material (left are real videos, right are reconstructions) to support our hypothesis further. The results demonstrate the high quality reconstructions from our autoencoder without blurry artifacts. Moreover, we think low reconstruction FVD (lower is better) that we provided in Table 4a and 4b implies high-quality video encoding hardly suffer from blurry artifact, because with such artifact, it is unlikely to achieve FVDs at our level asd due to the high distribution shift. Note that this metric has been widely used in previous latent video generation works [1, 2] to measure the reconstruction quality of videos.
>
> [1] Yu et al., MAGVIT: Masked Generative Video Transformer, CVPR 2023.
> [2] Yu et al., Video Probabilistic Diffusion Models in Projected Latent Space, CVPR 2023.

---

### Public Comment · ~Haonan_Qiu1 · 2023-11-18
**Code of Paper**

Thanks for your impressive work. The idea of decomposition is interesting. Do you plan to release your code (maybe only the code for inference) if the paper is accepted?

---

> ### Public Comment · ~Haonan_Qiu1 · 2023-11-20
>
> Meanwhile, using the average value of images seems not a good idea. For example in the uploaded example_videos.mp4, the panda case has significant movement however it brings some obvious artifacts around the panda. It may be better to represent content with some position insensitive embedded latent. That may also help to address the concern of Reviewer BXkg.

---

> > ### Author Response · Authors · 2023-11-22
> > **Response to Haonan Qiu**
> >
> > Thanks for an interest to our work! We respond to each of your comments one-by-one in what follows.
> >
> > **[Q1] Plan for releasing an implementation.**
> >
> > We are planning to release the implementation.
> >
> > **[Q2] Average value of images seems not a good idea.**
> >
> > Thanks for pointing this out. In principle, since our design includes not only a content frame but also a “motion latent vector”, the learned motion latent will compensate for the representing power of the content frame, and reconstruct each frame with a high quality. This has been empirically supported by the high-quality encoding performance in Table 4a and 4b. In practice, we think the appearance of blurriness in generated videos may be attributed to a relatively not large dataset and a relatively small model size. Hence, we believe visual quality can be much better if we use a larger text-video dataset, as validated in the concurrent work [1]. Moreover, recall that our model includes a “single” diffusion model that has a relatively small model size (1.7B including the autoencoder) in total without any cascaded modules, unlike the existing video diffusion models; we also believe training larger models will greatly improve the frame quality.
> >
> > [1] Wang et al., VideoFactory: Swap Attention in Spatiotemporal Diffusions for Text-to-Video Generation, arXiv 2023

---

### Author Response · Authors · 2023-11-22
**Common Response**

Dear reviewers and AC,

We sincerely appreciate your valuable time and effort spent reviewing our manuscript.

As reviewers highlighted, our paper is well-written (all reviewers), provides a novel (Reviewer jHjr, cm4G, BXkg), efficient (Reviewer jHjr, cm4G, BXkg) method for video generation, verified with strong performance (all reviewers). We believe our paper provides a new paradigm for training an efficient video diffusion model based on a new video encoding scheme via content-frame motion-latent decomposition.

We appreciate your valuable comments on our manuscript. In response to the questions and concerns you raised, we have carefully revised and improved the manuscript with the following additional discussions and experiments:

- Fix a typo and improve clarity (Abstract)
- Additional experiment on the WebVid-10M dataset with larger frame strides (Figure 1, Table 3, Appendix G)
- Added references with additional discussion on video generation with content/motion decomposition (Appendix A)
- Illustrations of generated videos with the same content vector and different motion latent vectors (Appendix F)
- Visualizations of reconstruction from our autoencoder (Appendix H, Supplementary Material)

These updates are temporarily highlighted in "blue” for your convenience to check.

We sincerely believe that CMD can be a useful addition to the video generation community, along with the above updates helping us better clarify the effectiveness of our method.

Thank you very much!
Authors.

---

### Meta-Review · Area_Chair_b5e2 · 2023-12-09

**Metareview:**

All the reviewers are positive of the paper: well-written and easy to follow, decomposing video generation into content and motion is effective, achieving SOTA results with low memory and computational costs while being able to utilize pretrained models. Meanwhile, there were also some notable concerns: the idea is not original as shown by similar methods, whether the autoencoder can learn meaningful motion representation, representing content as a weighted sum of frames may limit the model to capture key components of the video, handling longer videos, and needs for more comprehensive evaluations and comparisons.

The authors did a very good job addressing most of the concerns from the reviewers and hence it received two 6 and two 8 ratings. Hence the AC recommends accept, while recognizing that the high-level idea is not original and therefore the contribution is in implementation strategy. The generated videos look visually pleasing but also blurring especially around motion boundaries.

**Justification For Why Not Higher Score:**

The chance for this paper to be bumped to spotlight is slim because the high-level idea of decomposing video generation to frame and motion is not original (nothing is surprising) and the results are not mind blowing, especially considering the shocking qualities from startups e.g. Pika AI.

**Justification For Why Not Lower Score:**

The numbers are good; the visual quality is robust. The reviewers all recommended accept.

---

### Decision · Program_Chairs · 2024-01-16

Accept (poster)